# Atomic cobalt as an efficient electrocatalyst in sulfur cathodes for superior room-temperature sodium-sulfur batteries

Bin-Wei Zhang [1], Tian Sheng [2], Yun-Dan Liu[3], Yun-Xiao Wang [1], Lei Zhang[1], Wei-Hong Lai[1], Li Wang[1], Jianping Yang[4], Qin-Fen Gu[5], Shu-Lei Chou [1], Hua-Kun Liu [1] & Shi-Xue Dou [1]

The low-cost room-temperature sodium-sulfur battery system is arousing extensive interest owing to its promise for large-scale applications. Although significant efforts have been made, resolving low sulfur reaction activity and severe polysulfide dissolution remains challenging. Here, a sulfur host comprised of atomic cobalt-decorated hollow carbon nanospheres is synthesized to enhance sulfur reactivity and to electrocatalytically reduce polysulfide into the final product, sodium sulfide. The constructed sulfur cathode delivers an initial reversible capacity of 1081 mA h g$^{-1}$ with 64.7% sulfur utilization rate; significantly, the cell retained a high reversible capacity of 508 mA h g$^{-1}$ at 100 mA g$^{-1}$ after 600 cycles. An excellent rate capability is achieved with an average capacity of 220.3 mA h g$^{-1}$ at the high current density of 5 A g$^{-1}$. Moreover, the electrocatalytic effects of atomic cobalt are clearly evidenced by operando Raman spectroscopy, synchrotron X-ray diffraction, and density functional theory.

[1] Institute for Superconducting and Electronic Materials, Australian Institute of Innovative Materials, University of Wollongong, Innovation Campus, Squires Way, North Wollongong, NSW 2500, Australia. [2] College of Chemistry and Materials Science, Anhui Normal University, 241000 Wuhu, P.R. China. [3] Hunnan Key Laboratory of Micro-Nano Energy Materials and Devices, Xiangtan University, 411105 Hunan, P.R. China. [4] State Key Laboratory for Modification of Chemical Fibers and Polymer Materials, College of Materials Science and Engineering, Donghua University, 201620 Shanghai, P.R. China. [5] Australian Synchrotron, 800 Blackburn Road, Clayton, VIC 3168, Australia. Correspondence and requests for materials should be addressed to Y.-X.W. (email: yunxiao@uow.edu.au) or to S.-L.C. (email: shulei@uow.edu.au)

Currently, lithium-ion batteries (LIBs) play a dominant role in battery technologies for portable electronics because of their high capacity, high energy density, and reliable efficiency[1,2]. On the other hand, new emerging applications, such as electric vehicles and large-scale grids, require battery technologies with low costs and long cycle life[3–6]. Lithium-sulfur (Li/S) batteries have attracted intense attention due to high theoretical specific energy, environmental benignity, and the low cost and abundance of sulfur[7–9]. Due to efforts over decades, exciting progress on Li-S batteries has been achieved in terms of high capacity, prolonged service life, and remarkable rate capability, which are rapidly bringing this system near delivery to market. Meanwhile, it should be noted that the battery systems based on Li-ion storage are not suitable for large-scale applications, due to the high cost and insufficiency of Li resources[10,11]. Therefore, increasing interest is currently transferring to batteries based on low-cost and abundant sodium[12,13]. Room-temperature sodium-sulfur (RT-Na/S) batteries are among the ideal candidates to meet the scale and cost requirements of the market due to overwhelming advantages: a theoretical capacity of S ($1672 \, \mathrm{mA \, h \, g^{-1}}$), low cost, nontoxicity and resource abundance[14,15]. Nevertheless, RT-Na/S batteries, which share a similar reaction mechanism to the Li/S batteries, are facing critical problems with respect to low reversible capacity and fast capacity fade[16,17]. The poor conductivity of sulfur and sluggish reactivity of sulfur with sodium, resulting in a low utilization rate of sulfur and incomplete reduction to $Na_2S_x$ ($x \geq 2$) rather than complete reduction to $Na_2S$, are the main reasons for low accessible capacity. In addition, fast capacity fade during the charge–discharge progress occurs due to the dissolution of long-chain polysulfides in the electrolyte, which also leads to the rapid loss of active materials. Hence, effective materials design is the primary factor that is expected to improve the conductivity and activity of sulfur, and prevent the dissolution of polysulfides. So far, the reported sulfur hosts (for example, hollow carbon spheres[15], microporous carbon polyhedron sulfur composite[18], and conducting polymer[19]) could exhibit decent enhancement, but a huge leap is needed to reach the standard of practical applications. To the best of our knowledge, the best rate capacity and longest cycling stability for RT-Na/S batteries are observed in those containing the sulfur@interconnected mesoporous carbon hollow nanospheres (S@iMCHS) ($127 \, \mathrm{mA \, h \, g^{-1}}$ at $5 \, \mathrm{A \, g^{-1}}$)[20] and C-S polyacrylonitrile (c-PANS) ($150 \, \mathrm{mA \, h \, g^{-1}}$ after 500 cycles at $220 \, \mathrm{mA \, g^{-1}}$)[21], respectively. It is obvious that the sulfur cathodes based on traditional carbonaceous host materials are not capable of meeting the practical targets for large-scale RT-Na/S batteries.

Recently, novel sulfur hosts with inherent polarization, such as metallic oxides[22] and metal sulfides[23], have been investigated in Li/S cells. Compared with bare carbon materials, these polarized host materials have strong intrinsic sulfiphilic property, which are able to impede polysulfide dissolution due to the strong chemical interactions between the polar host materials and the polysulfides. A similar concept has been demonstrated in RT-Na/S batteries; Cu nanoparticles loaded in mesoporous carbon are utilized to immobilize the sulfur and polysulfides[24]; a novel Cu foam current collector is able to activate sulfur electroactivity as well[25]. Furthermore, atomic-scale metal materials, including single-atom metals and metal clusters, in general, not only possess amazing electronic and reactive properties, but also could reach the maximum atomic utilization[26–31]. It is rational but very challenging to introduce novel atomic metals into a sulfur host, which is expected to maximize the multifunctions of a polarized sulfur host and achieve extraordinary performance for RT-Na/S batteries.

Here, we successfully synthesized a highly effective sulfur host with atomic Co (including SA Co and Co clusters) supported in micropores of hollow carbon (HC) nanospheres. The HC nanospheres are employed as ideal frameworks, which could allow initial anchoring of Co nanoparticles and subsequent S encapsulation. In each HC reactor, it is interesting that the diffusion of sulfur molecules can serve as traction for atomic Co ($Co_n$) migration into carbon shells, forming a novel $Co_n$-HC host. A sulfur composite, sulfur encapsulated in a $Co_n$-HC host (S@$Co_n$-HC), is prepared by simply tuning the reaction temperature. When applied in RT-Na/S batteries, the S@$Co_n$-HC cathode exhibits outstanding electrochemical performance, which suggests that the maximized atomic utilization could optimize the multiple functions of Co metal towards enhancing sulfur conductivity, activating sulfur reactivity, and immobilizing sulfur and polysulfides. More specifically, the S@$Co_n$-HC achieves remarkable cycling stability ($507 \, \mathrm{mA \, h \, g^{-1}}$ after 600 cycles at $100 \, \mathrm{mA \, g^{-1}}$) and rate performance ($220.3 \, \mathrm{mA \, h \, g^{-1}}$ at $5 \, \mathrm{A \, g^{-1}}$). A deep insight into the mechanism has also been obtained by cyclic voltammetry (CV), operando Raman spectroscopy, synchrotron X-ray diffraction (XRD), and density functional theory (DFT), confirming that atomic Co could alleviate the "shuttle effect" and also effectively electrocatalyze the reduction from $Na_2S_4$ into the final product $Na_2S$.

## Results

**Growth process for sulfur-hosted atomic cobalt-decorated hollow carbon composite.** The synthetic process of the S@$Co_n$-HC is illustrated in Fig. 1. The successful encapsulation of Co nanoparticles (NPs, ~3 nm) and S is attributed to the microporous and hollow structure of carbon spheres. Initially, a $CoCl_2$ solution was immersed into the HC spheres and was reduced to Co NPs that uniformly decorated the carbon shells (~5 nm) of HC nanospheres (Co-HC) by controlled thermal treatment method (Supplementary Figs. 1, 2). The interactions between Co and S occur in two stages, along with increasing temperature. Firstly, the melted S was loaded into the Co-HC by a capillarity effect via a facile melt-diffusion strategy at 155 °C for 12 h (with the product denoted as S/Co-HC). It is clear that some of the S agglomerates in the hollow space of carbon spheres and others are dispersed in the carbon shells of the S/Co-HC, as shown using atomic resolution high-angle annular dark field (HAADF) scanning transmission electron microscopy (STEM) images (Supplementary Fig. 3). Subsequently, the S/Co-HC was heat-treated at 300 °C in a sealed quartz ampoule, which interestingly leads to the disappearance of Co nanoparticles and S agglomeration. During this process, S begins to sublime. The concentration gradient results in S diffusion from the inside of the nanospheres to the surface. With sufficient thermal energy for S evaporation, most of the S molecules diffuse into the C shells, which would drive the Co nanoparticles to be re-dispersed into the carbon shells as well. Thus, atomic Co, including Co single atoms and clusters, migrates into the C shells of each HC nanosphere by taking advantage of the diffusion of inner S molecules. Finally, a novel S nanocomposite with S embedded into atomic Co-decorated hollow carbon (S@$Co_n$-HC) could be achieved.

As displayed in Fig. 2 and Supplementary Fig. 4, the scanning electron microscopy (SEM) and transmission electron microscopy (TEM) images of the S@$Co_n$-HC demonstrate that the uniform dispersion of hollow carbons without any nanoparticles existed; meanwhile, atomic Co (bright dots) are observed in the C shells. The elemental mapping and line-profile analysis of S@$Co_n$-HC demonstrates that this atomic Co is well confined in the carbon shells; meanwhile, most of the S is embedded in the

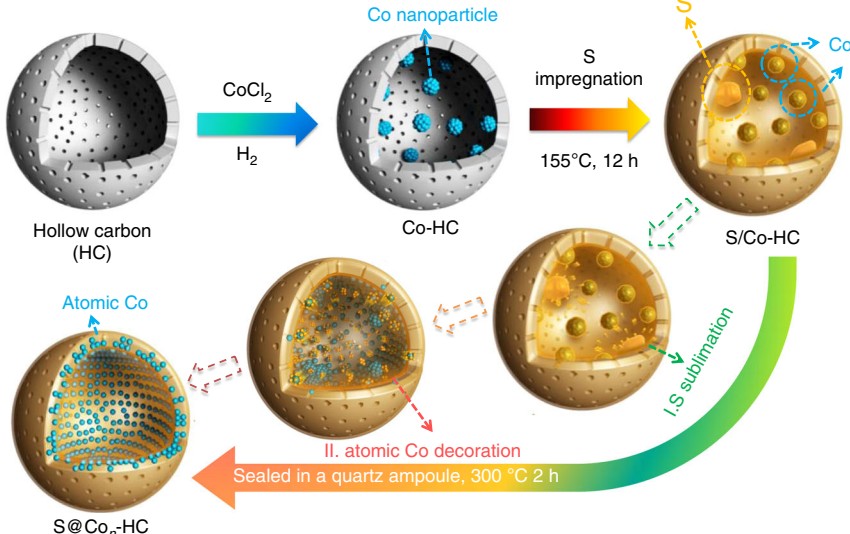

**Fig. 1** Schematic illustration of synthesis. Schematic illustration of the synthesis of the hollow carbon decorated with cobalt nanoparticles (Co-HC). After sulfur (S) impregnation, the S/Co-HC is heat treated to generate atomic Co-decorated hollow carbon as a sulfur host material (S@$Co_n$-HC)

carbon shell along with the dispersion of atomic Co, which implies the simultaneous formation of atomic Co and S dispersion. This is attributed to Co atoms migrating into HC shells with S sublimation via an atom migration strategy based on the strong interaction between Co and S. Hence, most of the S molecules diffuse into the C shells, and are adsorbed by atomic Co. The average size of the atomic Co is calculated to be 0.4 ± 0.2 nm from 200 single atoms and clusters in Supplementary Fig. 4. For comparison, a sample with S loading on plain HC (S@HC), in which the S is evenly dispersed among the carbon shells of HC, was prepared at 300 °C (Supplementary Fig. 5). It should be pointed out that atomic metals are difficult to form in pure carbon materials because of their high energy and instability[32]. Surprisingly, the atomic Co is successfully introduced into the S@$Co_n$-HC composite. Active S, in turn, plays a critical role in forming and stabilizing atomic Co by strong chemical Co−S bonds. In sharp contrast, numerous cubic nanoparticles (~ 10 nm) can be observed in HC prepared at 400 °C (Supplementary Fig. 6). The HAADF-STEM image displays two lattice distances of 1.94 Å and 2.75 Å, which are indexed to the (220) and (200) planes of $CoS_2$, respectively. Elemental mapping of S@$CoS_2$-HC clearly shows the formation of $CoS_2$. The line-profile analysis across the carbon shell in Supplementary Fig. 6c demonstrates that the signal of Co is negligible in the carbon shell. The elemental S mapping results demonstrate that the S is homogeneously dispersed in the $CoS_2$-HC host. Inductively coupled plasma-optical emission spectroscopy (ICP-OES) results demonstrate that the contents of Co are comparable, with weight ratios of 7.53, 7.06, and 6.85% in S/Co-HC, S@$Co_n$-HC, and S@$CoS_2$-HC, respectively. Meanwhile, the Co loading ratios (5 and 20% of $CoCl_2$) also have been optimized for S@Co-HC as shown in Supplementary Figs. 7, 8 (details see Supplementary Note 1).

The thermogravimetric analysis (TGA) results shown in Fig. 3a, Supplementary Figs. 9, 10 indicate that the S contents in S/Co-HC, S@$Co_n$-HC, and S@HC are ~48, 47, and 30 wt%, respectively. The low S loading ratio of 30 wt% indicates that atomic Co in HC is favorable to capture S and enhance S loading amount. There are three states of sulfur in S@$Co_n$-HC. The crystalline sulfur on the carbon layer would sublime at a relatively low temperature of ~270 °C, which accounts for ~33 wt%. Then, a small amount of amorphous sulfur, confined in the micropores[15],

would evaporate at temperatures from 270 to 530 °C with a sulfur loss of ~8 wt%; the sulfur encapsulated in the hollow space could finally sublime at a high temperature of 530 °C, which corresponds to a sulfur portion of ~6 wt%. The S@HC sample shows a similar TGA curve, indicating S present in the same states as those of the S@$Co_n$-HC; the amorphous sulfur in S@HC is about ~7 wt%. Compared with other Co-based materials, as shown in Supplementary Fig. 10, S in the S@$Co_n$-HC is the most difficult to vaporize. The starting temperature of weight loss is 173 °C for S@$Co_n$-HC, which is much higher than that of S/Co-HC (155 °C), indicating that the binding between S and Co in S@$Co_n$-HC is the strongest[20]. Interestingly, the S loss commences at 171 °C for S@HC, indicating that the S is firmly embedded into HC after removing the surface S via heat treatment at 300 °C[20]. This result also indicates that the S in S@$Co_n$-HC not only is physically confined in HC frameworks, but also chemisorbed by atomic Co. The S ratio of S@$CoS_2$-HC (~31 wt%) is low because the formation of $CoS_2$ consumes a certain amount of S. XRD patterns of these samples are shown in Fig. 3b and Supplementary Fig. 11; the peaks of S@$Co_n$-HC and S@HC are indexed to crystalline sulfur. The low intensity and absence of certain peaks imply that sulfur could be embedded in the $Co_n$-HC and HC hosts. $CoS_2$/S-HC has four peaks at 32.5°, 36.36°, 46.54°, and 54.98°, corresponding respectively to the (200), (210), (220), and (311) planes of $CoS_2$ (JCPDF no. 41-4171). Significantly, the XRD results for S/Co-HC and S@$Co_n$-HC indicated that S accounted for the dominant component, and the lack of XRD peaks for Co or any $CoS_x$ is likely due to the ultrafine and even atomic size of Co; additionally, the wrapping by S of the surface of Co would decrease its signal as well.

To investigate the interaction between Co and S, X-ray photoelectron spectroscopy (XPS) was carried out. As shown in Fig. 3c and Supplementary Fig. 12, compared with pure S (S $2p_{3/2}$,164.0 eV), the S $2p_{3/2}$ responses of S@HC and S@$Co_n$-HC are shifted at 163.60 and 163.45 eV, respectively. The shift is probably attributable to the adsorption of S by HC[33]. The lower S $2p_{3/2}$ of S@$Co_n$-HC could be due to the presence of atomic Co, which is decorated on the carbon shell and could aid HC in immobilizing S by forming Co−S bonds. Interestingly, the S $2p_{3/2}$ binding energy of S/Co-HC (165.1 eV) is close to that of $CoS_2$/S-HC (164.90 eV), which indicates that the surface Co nanoparticles of S/Co-HC could be polarized to $S^{2-}$. To further investigate this

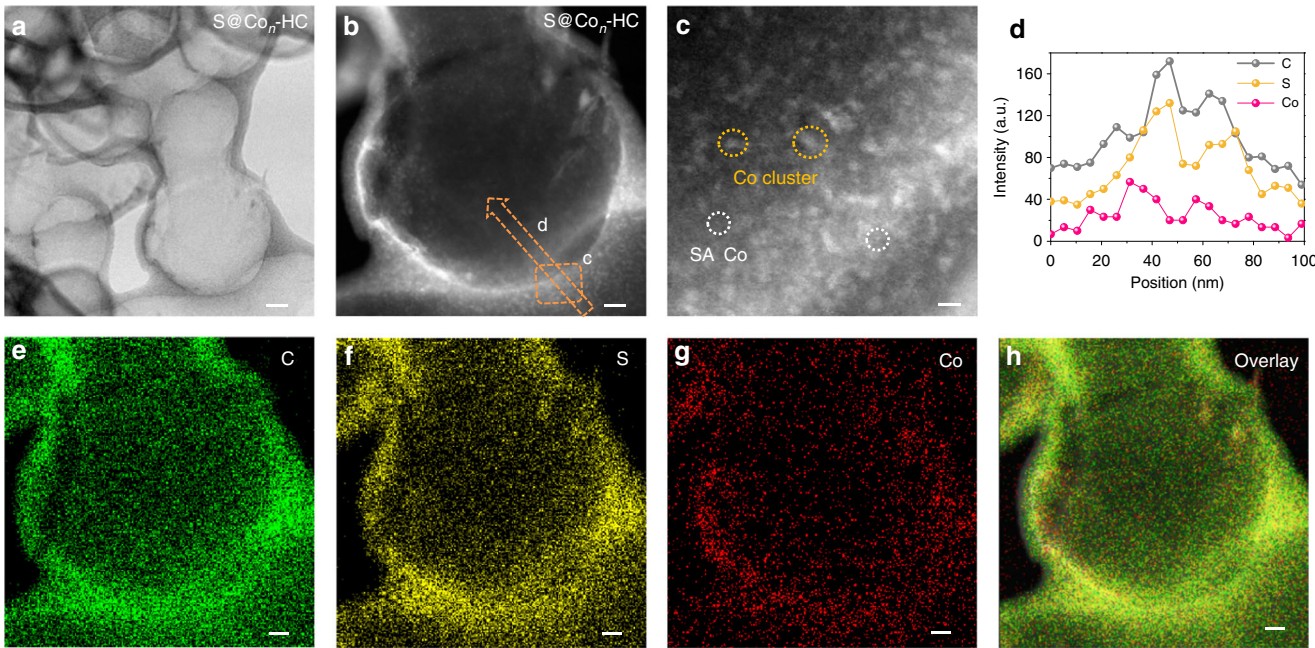

**Fig. 2** Representative electron microscopy images. **a** Transmission electron microscopy (TEM) image, **b**, **c** high-angle annular dark field (HADDF)-scanning tunneling electron microscopy (STEM) images of atomic cobalt-decorated hollow carbon sulfur host (S@Co$_n$-HC). Scale bar, 20 nm (**a**), 10 nm (**b**) and 2 nm (**c**). **d** Line-profile analysis from the area indicated on (**b**). **e–h** Elemental mapping of S@Co$_n$-HC. Scale bar, 10 nm

hypothesis, we studied the states of Co. The XPS data for S@Co$_n$-HC in the Co 2$p$ region in Fig. 3d indicate that Co contributions can be deconvoluted into Co$^0$ (778.70 eV) and Co$^{2+}$ (781.60 eV). The Co$^{2+}$ (781.60 eV) in S@Co$_n$-HC could be attributed to single Co anchored on S-dispersed hollow carbon[34], probably through the formation of a Co—S bond. While, except for single Co atoms, Co clusters exist in S@Co$_n$-HC as shown in Fig. 2 and Supplementary Fig. 4. Due to the existence of these Co clusters, XPS data in the Co 2$p$ region for S@Co$_n$-HC show evidence of the Co$^0$ state. The binding energy of the Co$^0$ 2$p_{3/2}$ in S@Co$_n$-HC is 778.70 eV, which is a shift of 0.5 eV compared with that of pure Co (778.20 eV); this right-shifted binding energy indicates the formation of Co—S bonds between Co clusters and S in S@Co$_n$-HC. The XPS spectrum region of Co 2$p_{3/2}$ for S@CoS$_2$-HC with peaks at 781.10 and 785.80 eV is attributed to Co$^{2+}$ 2$p_{3/2}$ and Co$^{4+}$ 2$p_{3/2}$, respectively, and the formation of Co—S bonds of CoS$_2$[35]. Since XPS analysis is a surface-sensitive technique, the trend in Co binding energy relies on the size of the Co@CoS$_x$ core-shell structure, and that is why the Co oxidation states of S/Co-HC show the highest bonding energy in Supplementary Fig. 12. Based on the TGA, XRD, and XPS results we could draw the conclusion for S@Co$_n$-HC that S is not only physically adsorbed by HC, but is also chemisorbed by atomic Co, leading to the formation of Co—S bonds. Meanwhile, the S@Co$_n$-HC delivers the Co$^0$ state, which could effectively improve conductivity of an S cathode and enhance the performance of RT-Na/S batteries.

**Performance evaluation of the room-temperature sodium-sulfur batteries**. The discharge/charge profiles of the 1st, 2nd, 10th, 50th, 100th, 200th, 300th, 400th, 500th, and 600th cycles at 100 mA g$^{-1}$ of S@Co$_n$-HC and S@HC cathode materials are shown in Fig. 4a, b. The RT-Na/S@Co$_n$-HC cell shows two long plateaus that run from 1.68 to 1.04 V, and 1.04 to 0.8 V during the initial discharge process: the high-voltage plateau corresponds to the solid—liquid transition from S to dissolved long-chain polysulfides; and the low-voltage plateau is attributed to the further sodiation of long-chain polysulfides to short-chain sulfides. By contrast, the two plateaus of S@HC are at 1.82 and 1.62 V during

the initial discharge process. The lower potential plateaus of S@Co$_n$-HC in the initial cycle may be attributed to the complex bonds between Co and S (Co—S bonds), so that additional energy is needed to dissociate S from the Co—S bond, resulting in a more negative potential[36,37]. Consequently, the following discharge potential plateaus of S@Co$_n$-HC shifted to the positive direction[38,39]. This phenomenon also could be found in S/Co-HC and S@CoS$_2$-HC, as shown in Supplementary Fig. 13. To investigate the effects of slow charge—discharge processes, the S@Co$_n$-HC cell at low current densities (20 and 50 mA g$^{-1}$) were carried out, as shown in Supplementary Fig. 14. It could be clearly seen that the initial reversible capacity of S@Co$_n$-HC is 1613 mA h g$^{-1}$ at 20 mA g$^{-1}$, which is close to the theoretical capacity of S (1672 mA h g$^{-1}$), retaining reversible capacity of 945 mA h g$^{-1}$ after 40 cycles. When tested at 50 mA g$^{-1}$, the S@Co$_n$-HC delivers an initial reversible capacity of 1360 mA h g$^{-1}$, maintaining 904 mA h g$^{-1}$ after 40 cycles. During the slow charge—discharge process at current density of 20 mA g$^{-1}$, the produced long-chain polysulfides could be further fully sodiated to Na$_2$S$_4$. Meanwhile, the atomic Co will effectively alleviate dissolution of Na$_2$S$_4$ and electrocatalytically reduce Na$_2$S$_4$ into the final product Na$_2$S. However, the slow charge—discharge process would aggravate the dissolution and shuttle effect of the long-chain polysulfides, leading to fast capacity decay and inferior capacity retention. This phenomenon is well in agreement with the cycling performance, in which this cathode shows the lowest capacity retention (58.5%) at 20 mA g$^{-1}$. The comparisons at different currents indicate that the slow charge—discharge process is favorable to realize high reversible capacity but severe capacity decay. It is rational to select a current density that would be slow enough to exert the capacity of all S active materials and fast enough to alleviate the shuttle effect. By contrast, the current density of 100 mA g$^{-1}$ shows the most satisfactory performance. Meanwhile, the electrochemical performances of different Co loading of S@Co-HC are shown in Supplementary Fig. 15 and Supplementary Note 2, which also demonstrated that the S@Co$_n$-HC processes the best performance among these cathode materials.

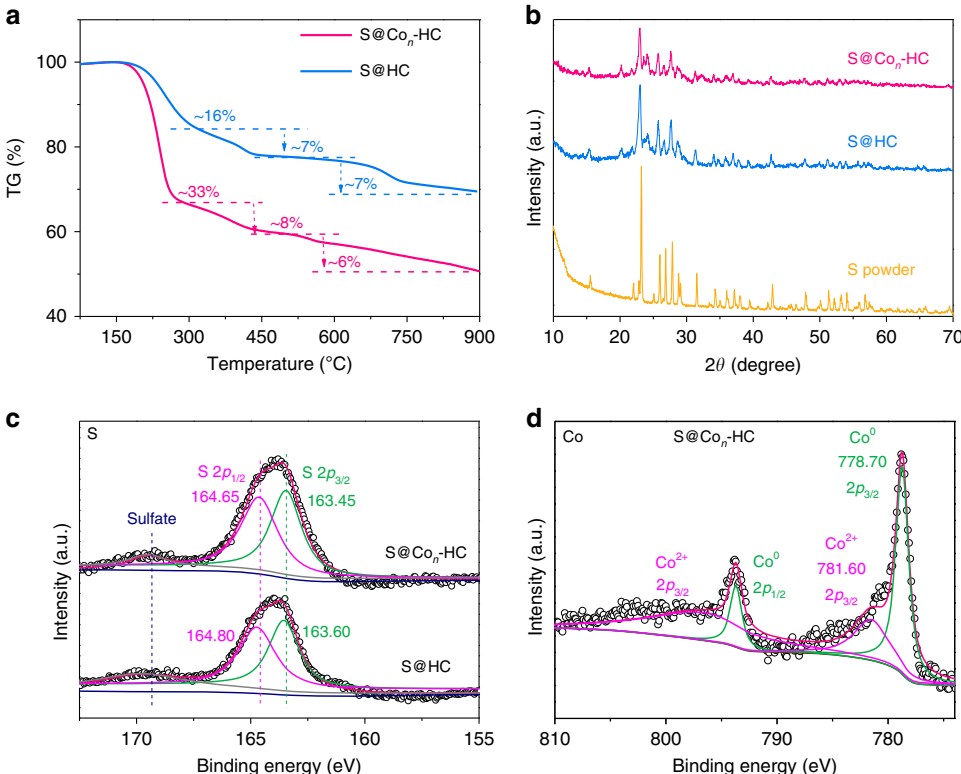

**Fig. 3** Thermogravimetric analysis, X-ray diffraction, and X-ray photoelectron spectra. **a** Thermogravimetry (TG) of hollow carbon hosting sulfur (S@HC) and atomic cobalt-decorated hollow carbon sulfur host (S@Co$_n$-HC). **b** X-ray diffraction (XRD) patterns of sulfur (S) powder, S@HC and S@Co$_n$-HC. **c** S 2$p$ region of X-ray photoelectron spectroscopy (XPS) spectra for S@HC (bottom) and S@Co$_n$-HC (top). **d** Co 2$p$ region of XPS spectrum for S@Co$_n$-HC

The long-term cycling stability of the S@HC and S@Co$_n$-HC cathodes is displayed in Fig. 4c at 100 mA g$^{-1}$ over 600 cycles. Both S@HC and S@Co$_n$-HC display high cycling stability and capacity retention after the initial capacity decay, which indicates that the closed hollow carbon host could effectively manage the fatal polysulfide dissolution. The S@$_{Con}$-HC delivers an initial reversible capacity of 1081 mA h g$^{-1}$ with a Coulombic efficiency of 52.1%, retaining excellent reversible capacity of 508 mA h g$^{-1}$ after 600 cycles. The high initial discharge capacity of S@Co$_n$-HC (~2075 mA h g$^{-1}$) is due to the decomposition of the electrolyte, the side reactions between the carbonate-based solvents and soluble polysulfides, and the formation of the solid electrolyte interphase film[25]. In sharp contrast, the S@HC cathode delivers the first capacity of 580/1209 mA h g$^{-1}$, which declines to 271 mA h g$^{-1}$ after 600 cycles. During the first ten cycles, there is obvious capacity decay for both of the S@Co$_n$-HC and S@HC cathodes, which is attributed to the loss of dissolved long-chain polysulfides. The cells show relatively stable cycling but with gradual capacity loss for the subsequent 600 cycles, which mainly originates from the impedance increase in the cells due to the formation of Na$_2$S. This is consistent with the synchrotron XRD results (Fig. 5), confirming that the nonconductive Na$_2$S would accumulate in the cathode during the charge/discharge processes. Significantly, the high accessible capacity of S@Co$_n$-HC arises mostly due to the atomic Co decoration that is able to further improve the conductivity and electroactivity of S. To highlight the role of atomic Co, the cycling stability of S/Co-HC and S@CoS$_2$-HC are shown in Supplementary Fig. 16. It is noteworthy that the S/Co-HC displays fast capacity degradation, which shows the initial reversible capacity of 1018/617 mA h g$^{-1}$, but after 100 cycles, it is only 64/62 mA h g$^{-1}$. Additionally, the first-cycle reversible capacity of S@CoS$_2$-HC is 610/1415 mA h g$^{-1}$ respectively; after 200 cycles, it is only 206 mA h g$^{-1}$. These results

demonstrate that the atomic Co possesses stronger electrocatalytic capability than Co nanoparticles and CoS$_2$ nanoparticles. The role of atomic Co towards improving S performance will be discussed in the following section.

Rate-capability tests were evaluated at various current densities from 0.1 to 5 A g$^{-1}$ in the potential range of 0.8 to 2.8 V, as shown in Fig. 4d. It is evident that S@Co$_n$-HC exhibits the highest reversible capabilities of ~820, 498, 383, 313, 269, and 220 mA h g$^{-1}$ at 0.1, 0.2, 0.5, 1, 2, and 5 A g$^{-1}$, respectively, compared to the S@HC and S@CoS$_2$-HC (Supplementary Fig. 17). When the discharge/charge rate is brought back to the initial rate of 0.1 A g$^{-1}$, RT-Na/S@Co$_n$-HC shows amazing reversible capacity of 625 mA h g$^{-1}$ after 100 cycles (367 mA h g$^{-1}$ for RT-Na/S@HC). A comparison of the rate capability versus current density of S@Co$_n$-HC with the state-of-the-art in the literature is presented in Fig. 4e; to the best of our knowledge, such an exceedingly high rate capability of RT-Na/S batteries has not been reported previously [39–45]. The polarized Co$_n$-HC host is responsible for the prevailing Na-storage properties of S@Co$_n$-HC, which plays key roles in maximizing sulfur/polysulfides immobilization and activation via strong electrocatalytic atomic Co, reaching performance that is among the best in the field of RT-Na/S batteries.

**Mechanistic investigation on sodium-storage of the sulfur cathode.** To investigate the mechanism of S@Co$_n$-HC, CV, in situ Raman spectroscopy (at 500 mA g$^{-1}$) and in situ synchrotron XRD ($\lambda = 0.6883$ Å) data, using the Powder Diffraction Beamline (Australian Synchrotron), were collected for the initial galvanostatic charge/discharge and the second discharge curve (at 100 mA g$^{-1}$). Figure 5a presents cyclic voltammograms of S@Co$_n$-HC, while voltammograms for S@Co-HC, S@CoS$_2$-HC, and

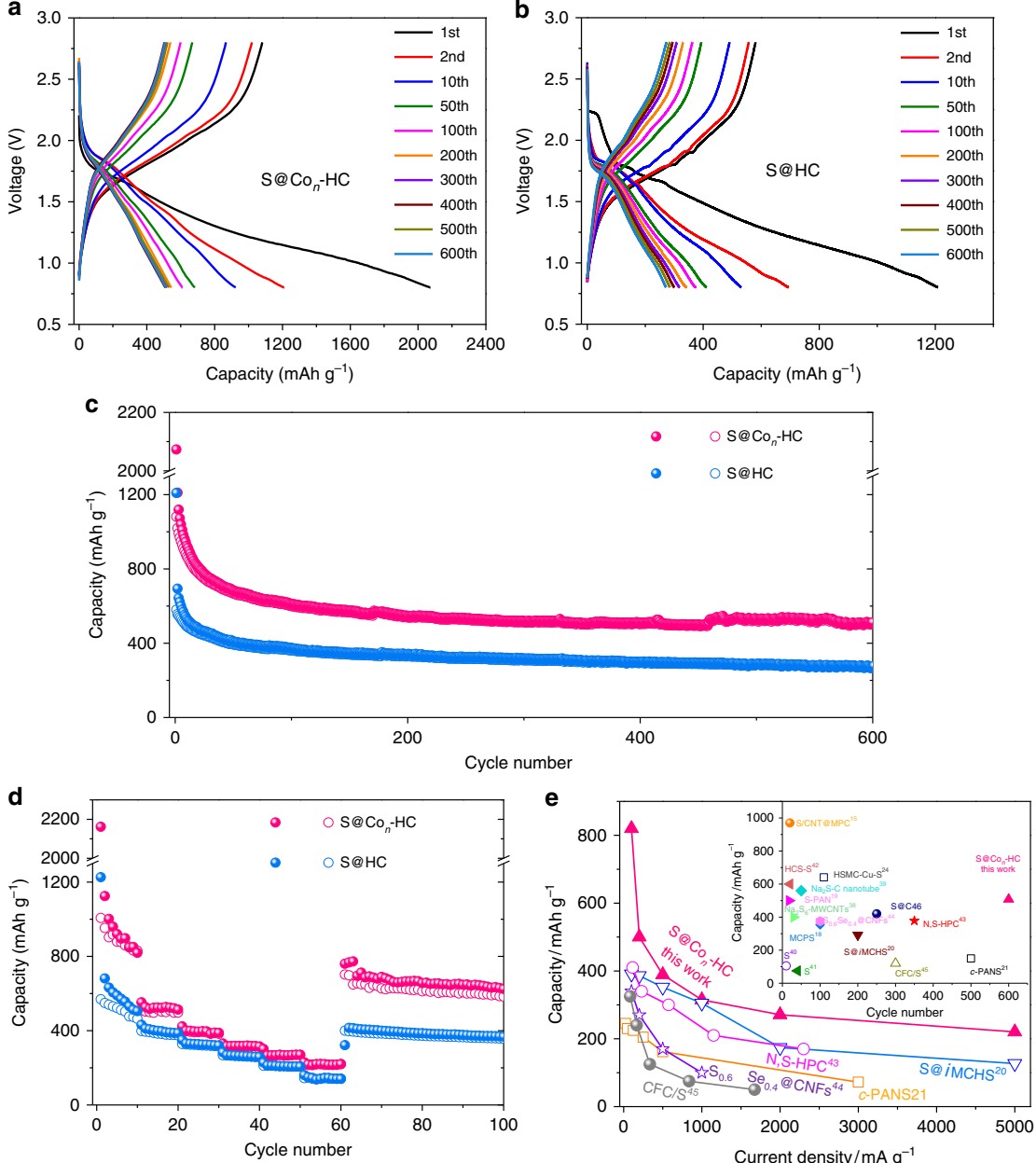

**Fig. 4** Room-temperature sodium-sulfur battery test. **a**, **b** Discharge/charge curves of atomic cobalt-decorated hollow carbon sulfur host (S@Co$_n$-HC) and hollow carbon hosting sulfur (S@HC) at 100 mA g$^{-1}$. **c**, **d** Cycling performance and rate performance for S@Co$_n$-HC and S@HC. **e** Comparison of the rate and cycling (inset) capabilities of previously reported room-temperature sodium-sulfur (RT-Na/S) batteries with our work

S@HC are shown in Supplementary Fig. 18 (details see Supplementary Note 3). The RT-Na/S@Co$_n$-HC cell shows two prominent peaks at around 1.68 and 1.04 V during the first cathodic scan. The peak at 1.68 V corresponds to the transition from solid S to dissolved liquid long-chain polysulfides (Na$_2$S$_x$, $4 < x \leq 8$)[46]; in the following cathodic sweep from 1.68 to 1.04 V, the long-chain polysulfides are further sodiated to Na$_2$S$_4$ and then short-chain polysulfides are sodiated (Na$_2$S$_y$, $1 < y \leq 3$)[20]. Significantly, the following cathodic peaks move toward positive potential after the first CV cycle, corresponding to the results for the discharge/charge curves, which also demonstrates the formation of Co−S bonds in S@Co$_n$-HC. Meanwhile, operando Raman spectra and synchrotron XRD patterns complementarily confirm the mechanism mentioned above. As illustrated in Fig. 5b, when the cell is discharged to 1.60 V, the S stretching vibration band at 475 cm$^{-1}$ disappears and another peak (451 cm$^{-1}$) appears, which

could be assigned to Na$_2$S$_4$[47]. Correspondingly, in situ synchrotron XRD (Fig. 5c) demonstrates broadening of a peak at 23.01°, indexed to the (240) planes of S (JCPDF no. 71-0569), upon discharge to 1.8 V. A new peak (22.97°) evolves around the original peak (23.01°), which could be attributed to the formation of long-chain polysulfides (Na$_2$S$_x$). When further discharged to 1.4 V, the Na$_2$S$_x$ peak gradually disappeared and a new peak at 13.22° developed, which can be attributed to the (213) planes of Na$_2$S$_4$ (JCPDF no. 71-0516). When discharged to 1.30 V, not only is there a main broad band at 451 cm$^{-1}$, but also a new peak at 472 cm$^{-1}$ that appears in the Raman spectra; this new peak could be attributed to the Na$_2$S$_2$[47]. Consistently, a new peak at 18.73° in the synchrotron XRD pattern for the sample discharged to 1.2 V could be attributed to the (104) peak of Na$_2$S$_2$ (JCPDF no. 81-1764)[20]. Furthermore, the in situ Raman spectrum of S@Co$_n$-HC that is discharged to 1.0 V also exhibits a new peak at 475 cm$^{-1}$.

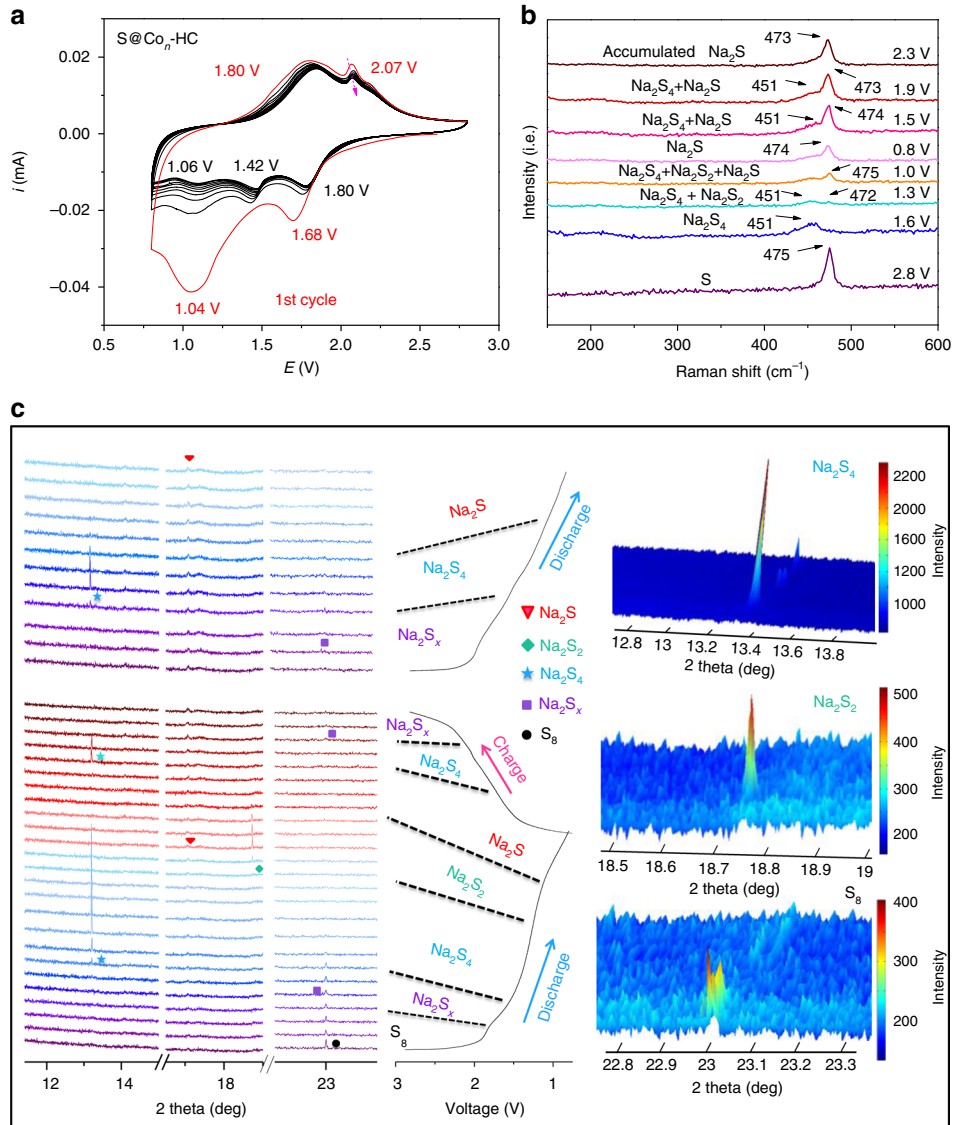

**Fig. 5** Characterization of mechanism. **a** Cyclic voltammograms, and **b** in situ Raman spectra. **c** In situ synchrotron X-ray diffraction (XRD) patterns of the room-temperature sodium-sulfur battery comprised of atomic cobalt-decorated hollow carbon sulfur host (RT-Na/S@Co$_n$-HC) cells (left) with the initial galvanostatic charge/discharge curves (middle) at 500 mA g$^{-1}$, and contour plot of XRD patterns at selected ranges of degrees two theta (right) at 100 mA g$^{-1}$

Given the similar Raman fringes of Na$_2$S and S$_8$[47,48], it mostly indicates the formation of Na$_2$S[47]; when fully discharged to 0.8 V, the only band at 475 cm$^{-1}$ demonstrates that the final product is Na$_2$S. It is convincing that a new peak generated at 17.07° could be assigned to the (220) planes of Na$_2$S as well, as shown in Fig. 5c (JCPDF no. 77-2149)[20]. Therefore, the first discharge mechanism is proposed to be as follows:

$$S \rightarrow Na_2S_x \rightarrow Na_2S_4 \rightarrow Na_2S_2 \rightarrow Na_2S. \tag{1}$$

When the cell is charged back to 2.8 V, Na$_2$S$_2$ and S are not detectable by in situ Raman spectroscopy or in situ synchrotron XRD, indicating that the reaction is not (or is only slightly) reversible; the processes from Na$_2$S to Na$_2$S$_4$ and to Na$_2$S$_x$ are expected to be reversible. The peaks corresponding to Na$_2$S in the Raman spectra and in the synchrotron XRD patterns always exist after its initial generation, which is probably due to the partial reversibility of the final Na$_2$S product, thus accumulating during the prolonged discharge/charge process.

Significantly, synchrotron XRD data for the second discharge process do not show any trace of Na$_2$S$_2$, and the diffraction peak intensity of Na$_2$S$_4$ obviously decreases. It indicates that the reaction rate of reduction from Na$_2$S$_4$ into Na$_2$S is very fast. We thoroughly analyzed this phenomenon, and proposed a new mechanism in which atomic Co could quickly catalyze the reduction of Na$_2$S$_4$ into Na$_2$S; this electrocatalytic reaction could effectively slow down the dissolution of Na$_2$S$_4$ during cycling as well as result in the excellent electrochemical performance of S@Co$_n$-HC. Furthermore, the polysulfide dissolution behaviors of S@Co$_n$-HC and S@HC electrodes using transparent glass cells are compared in Supplementary Fig. 19. The cell with S@Co$_n$-HC remained colorless during the 10-h discharge process, which implies the alleviation of the polysulfide dissolution and suggests that atomic Co could kinetically catalyze the polysulfide reduction to Na$_2$S instead of dissolution into the electrolyte. However, the yellow polysulfide on the surface of the S@HC

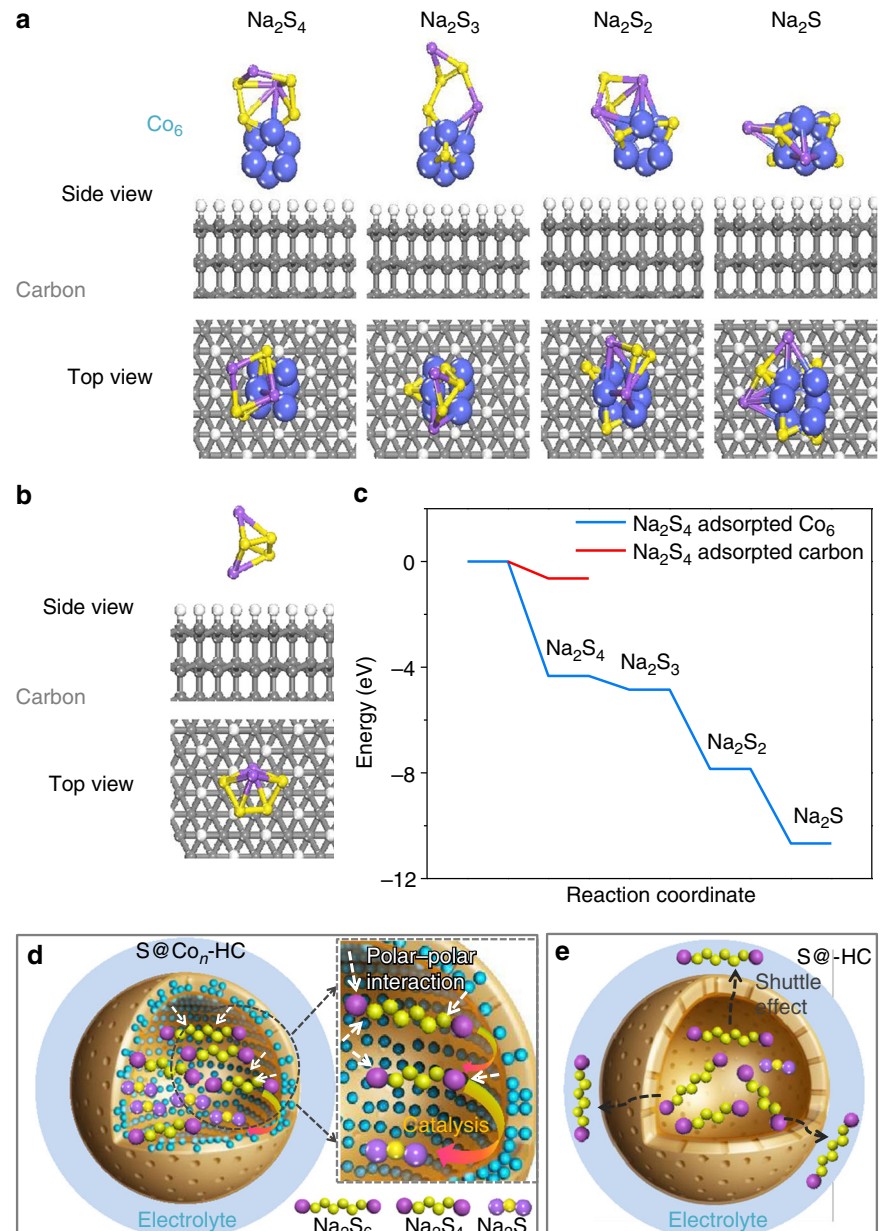

**Fig. 6** Density functional theory results and electrode reaction mechanism. **a** Optimized structures of $Na_2S_4$ cluster on carbon-supported $Co_6$ cluster, and **b** on carbon support. Purple: Na; yellow: S; blue: Co; gray: C; white: H. **c** Energy profiles of $Na_2S_4$ adsorption on carbon-supported $Co_6$ cluster (in blue) and carbon support (in red). **d**, **e** Schematic illustrations of electrode reaction mechanism of atomic cobalt-decorated hollow carbon sulfur host (S@$Co_n$-HC) and hollow carbon hosting sulfur (S@HC), respectively

electrode was observed upon discharge for 5 h; when upon a 10-h sodiation process, it could be clearly seen that yellow polysulfide dissolved in the cell. This color change of S@HC indicates that the polysulfide dissolution into electrolyte, i.e. shuttle effect, could lead to a loss of active materials. In order to guarantee reliability of the capacity of the S@$Co_n$-HC cathode, the capacity contribution of the S host, $Co_n$-HC, was evaluated as well. The $Co_n$-HC was fabricated from the S@$Co_n$-HC sample by dissolving the loaded S with $CS_2$ solvent. The XRD results of $Co_n$-HC and S@$Co_n$-HC are shown in Supplementary Fig. 20. It could be clearly seen that $Co_n$-HC does not show any S characteristic peaks, indicating that S has been completely removed. The discharge/charge profiles and cycling performance of $Co_n$-HC are shown in Supplementary Fig. 21a, which displays a very low initial reversible capacity of 70 mA h g$^{-1}$, only retaining a

reversible capacity of 40.1 mA h g$^{-1}$ after 200 cycles. By contrast, Supplementary Fig. 21b clearly shows that the capacity contribution of $Co_n$-HC in the S@$Co_n$-HC cathode could be negligible. Meanwhile, the compositional and morphological changes of S@$Co_n$-HC after 600 cycles are shown in Supplementary Fig. 22, which also indicated that the atomic Co in S@$Co_n$-HC could effectively enhance the reversible capacity of the RT-Na/S@$Co_n$-HC batteries.

In order to confirm our hypothesis, ab initio molecular dynamics (AIMD) simulations are used to reveal the decomposition of the $Na_2S_4$ cluster adsorption process on atomic Co/carbon (Fig. 6a) and carbon support (Fig. 6b). Figure 6a, b shows the decomposition of $Na_2S_4$ cluster and evolution into $Na_2S_3$ cluster, $Na_2S_2$ cluster, and $Na_2S$ cluster on atomic Co/carbon and carbon support. An ideal model of a $sp^3$ carbon, including 216 C atoms

and two exposed surfaces terminated by 72 H atoms[49], is applied in modeling the carbon support to calculate the adsorption of a $Na_2S_4$ cluster. The DFT calculations were conducted by considering the single atomic Co occupying 41% in the S@$_{Con}$-HC and the $Co_6$ cluster consisting of six Co atoms with the size of ~0.1 nm. The adsorption energy was defined as: $E(ad) = E(ad/surf) - E(surf) - E(ad)$, where $E(ad/surf)$, $E(surf)$, and $E(ad)$ are the total energies of the adsorbates binding to surface, clean surface and free adsorbate in gas phase, respectively. The adsorption energy of $Na_2S_4$ cluster on carbon support is −0.64 eV. The binding energy of the $Co_6$ cluster with the carbon support layer is −1.21 eV; meanwhile, the $Na_2S_4$ initially adsorb on the $Co_6$ cluster with the binding energy of −0.64 eV, which is the same with that on the $sp^3$ carbon surface. However, the $Na_2S_4$ structure was observed to decompose spontaneously on the $Co_6$ cluster during the AIMD simulation; for pure carbon support, $Na_2S_4$ could not be decomposed. As presented in Fig. 6a, $Na_2S_3$, $Na_2S_2$, and $Na_2S$ clusters were identified respectively on the $Co_6$ cluster and the dissociated S atoms were trapped by the $Co_6$ cluster. Figure 6c displays the relative adsorption energies of these sodium polysulfide clusters and the corresponding data are listed in Supplementary Table 1, showing that the adsorption energy of $Na_2S_4$ on $Co_6$ is −4.33 eV; for $Na_2S_3$, the adsorption energy is negatively shifted to −4.85 eV. Furthermore, the adsorption energy of $Na_2S_2$ is −7.85 eV; surprisingly, the adsorption energy of $Na_2S$ negatively shifts to −10.67 eV. This strong adsorption energy of $Na_2S$ indicates that the reaction from $Na_4S_2$ into $Na_2S$ is kinetically fast. It is evident that the binding energies of these sodium polysulfide clusters were much stronger than those on pure carbon support, indicating that the decomposition of $Na_2S_4$ in the presence of the $Co_6$ cluster could be electrocatalyzed, consistent with the speculation from operando Raman and synchrotron XRD results. The schematic illustrations of electrode reaction mechanisms for the S@$Co_n$-HC and S@HC are shown in Fig. 6d, e. These atomic Co, with surface sulfurization, could effectively alleviate the polysulfides dissolution based on polar−polar interactions. Moreover, the confined polysulfides in the inner carbon shell could be fully catalytically reduced into $Na_2S$ by atomic Co, leading to high S utilization. Therefore, the atomic Co in S@$Co_n$-HC plays a critical role in achieving sustainable cycling stability and high reversible capacity. By contrast, the intensive "shuttle effect" and incomplete sodiation reactions result in the inferior performance of the S@HC cathode.

## Discussion

Overall, atomic Co, including SA Co and Co clusters, is successfully applied into RT-Na/S batteries as a superior electrocatalytic host. The novel S@$Co_n$-HC electrode delivers a high initial reversible capacity of 1081 mA h g$^{-1}$; even after 600 cycles, it achieves a superior reversible capacity of 508 mA h g$^{-1}$ at 100 mA g$^{-1}$ without any degeneration of the elaborate nanostructure. The atomic scale of polarized Co is responsible for the outstanding enhancement of the S cathode, which is reaching the limitation of Co (Co-S) for S/polysulfides immobilization and activation in RT-Na/S batteries. Meanwhile, in situ Raman, synchrotron XRD, and DFT are combined to confirm that atomic Co could electrocatalytically reduce $Na_2S_4$ into $Na_2S$, which effectively alleviates dissolution of polysulfides and thus impeding the shuttle effect. Significantly, this work introduces atomic Co into electrode design, which innovatively bridges battery and electrocatalyst fields and provides a new exploration direction for novel design of electrode materials for the advancement of various battery technologies, especially in RT-Na/S batteries.

## Methods

**Synthesis of hollow carbon nanospheres.** Commercial silicon nanoparticles (~60–70 nm), utilized as hard templates, were first coated with resorcinol formaldehyde (RF) via a sol−gel process. Specifically, 0.15 g Si nanoparticles and 0.46 g cetyltrimethylammonium bromide (CTAB) were added in 14.08 mL of $H_2O$ and transferred into a three-neck round-bottom flask. A homogenous dispersion could be obtained after continuous ultrasonication and stirring for 0.5 h, respectively. Secondly, 0.7 g resorcinol, 56.4 mL of absolute ethanol, and 0.2 mL of $NH_4OH$ were added in the dispersion sequentially; the flask was maintained at 35 °C with stirring for 0.5 h, followed by the addition of 0.1 mL formalin. The RF polymerization could be completed after continually stirring for 6 h at 35 °C and ageing overnight. The obtained Si@RF nanospheres were collected and washed with deionized water and alcohol, respectively. The core-shell Si@C sample was prepared by calcination of the Si@RF powder at 600 °C for 4 h (5 °C min$^{-1}$) in $N_2$ atmosphere. Finally, hollow carbon nanospheres (HC) were prepared by etching the Si template away with a 2.0 M NaOH solution.

**Synthesis of different sulfur cathode samples.** A sulfur host, cobalt nanoparticles-decorated HC (Co-HC), was synthesized by uniform dispersion of 44.76 mg $CoCl_2$ and 100 mg HC in ethanol via ultrasonication. The HC containing $CoCl_2$ was then heated overnight in a blast oven at 80 °C, by which the mixture could solidify and shrink along with the ethanol evaporation. Afterwards, the above mixture was reduced at 200 °C for 2 h in a forming gas with 10 vol% $H_2$ in nitrogen, leading to the formation of Co-HC. Three S cathode samples were fabricated accordingly based on this Co-HC host. A mixture of Co-HC:S with a weight ratio of 1:1.5 was first ground by mortar and pestle, and then sealed in a Teflon-lined autoclave. A primary S cathode, S/Co-HC composite, was obtained after the autoclave was heated at 155 °C for 12 h. When the obtained S/Co-HC composite was further sealed in a quartz ampoule, and thermally treated at 300 and 400 °C for 2 h in $N_2$ atmosphere, respectively, two new samples denoted as S@$Co_n$-HC and S@$CoS_2$-HC could be synthesized. In addition, a contrast sample with plain HC as S host was prepared, in which S was embedded into the plain HC frameworks (denoted as S@HC). The synthesis procedures are the same as that of S@$Co_n$-HC by utilizing HC instead of Co-HC.

**Structural characterization.** The morphologies of the samples were investigated by SEM (JEOL 7500), TEM (JEOL 2011, 200 keV), and STEM (JEOL ARM-200F, 200 keV). The XRD patterns were collected by powder XRD (GBC MMA diffractometer) with Cu Kα radiation at a scan rate of 1° min$^{-1}$. XPS measurements were carried out using Al Kα radiation and fixed analyzer transmission mode: the pass energy was 60 eV for the survey spectra and 20 eV for the specific elements.

**Electrochemical measurements.** The electrochemical tests were conducted by assembling coin-type half-cells in an argon-filled glove box. The slurry was prepared by fully mixing 70 wt% active materials (S/Co-HC, S@$Co_n$-HC, S@$CoS_2$-HC, S@HC), 10 wt% carbon black, and 20 wt% carboxymethyl cellulose (CMC) in an appropriate amount of water via a planetary mixer (KK-250S). Then, the obtained slurry was pasted on Cu foil using a doctor blade with a thickness of 100 μm, which was followed by drying at 50 °C in a vacuum oven overnight. The working electrode was prepared by punching the electrode film into discs of 0.97 cm diameter. The sodium foil was employed as both reference and counter electrode. The electrodes were separated by a glass fiber separator. Electrolyte, 1.0 M $NaClO_4$ in propylene carbonate/ethylene carbonate with a volume ratio of 1:1 and 5 wt% fluoroethylene carbonate additive (PC/EC + 5 wt% FEC), was prepared and used in this work. The electrochemical performance was tested on a LAND Battery Tester with a voltage window of 0.8–2.8 V. All the capacities of cells have been normalized based on the weight of sulfur. CV was performed using a Biologic VMP-3 electrochemical workstation.

**In situ measurements.** The in situ Raman cell was bought from Shenzhen Kejing star. The in situ Raman was collected with a Renishaw InVia Raman microscope, with excitation 532 nm laser wavelengths and L50× objective lens. The spectra were collected in galvanostatic mode when the in situ Raman cell was discharged/charged at a current rate of 500 mA g$^{-1}$ using a computer controller (CHI 660D). The acquisition time of each Raman spectrum was 60 s; and lower laser power was utilized to avoid electrode damage during the long-term measurements. For in situ synchrotron XRD measurements, the cells were similar to the above-mentioned coin cells for electrochemical performance testing. To enhance the diffraction peak intensity, a thicker layer of cathode material was loaded on the Cu foil, with loading up to 5 mg cm$^{-2}$. To guarantee that the X-ray beams could penetrate the whole cell and that the electrochemical reactions could be monitored, three 4-mm diameter holes were punched in the negative and positive caps as well as the spacer. Then, Kapton film (only showing low-intensity responses in XRD patterns) was used to cover the holes in the negative and positive caps, and AB glue was used for complete sealing. The charge/discharge process was conducted with a battery test system (Neware) that was connected to the cell.

**Computational methods.** The spin-polarized electronic structure calculations were performed in the Vienna Ab-initio Simulation Package code with Perdew-Burke-

Ernzerhof (PBE) functional of exchange-correlation. The projector-augmented-wave (PAW) pseudopotentials were utilized to describe core electron interactions[50–52]. Considering the significance of van der Waals (vdW) forces to the adsorption, we utilized the D3 dispersion vdW corrections with zero damping for describing the vdW interactions.[53,54] The Co cluster consisted of six Co atoms with a size of ~0.1 nm and the Co−Co bond distances was 2.24 Å. The $Na_2S_4$ cluster was obtained after 10 ps of AIMD simulations at 350 K at first and the final structure was optimized. To gain insights into the $Na_2S_4$ dissociative adsorption on carbon-supported $Co_6$ cluster, we firstly performed the AIMD simulation for 10 ps (10,000 steps, 1 fs per step) within the canonical (NVT) ensemble at 350 K to accelerate the dissociation rate of $Na_2S_4$ cluster on carbon-supported $Co_6$ cluster. During the AIMD simulations, the carbon support was fixed while the $Co_6$ and $Na_2S_4$ clusters were allowed to move. Secondly, we chose some representative sodium polysulfide structures, i.e., $Na_2S_3$, $Na_2S_2$ and $Na_2S$ clusters, which were observed from molecular dynamics simulations. Thirdly, the geometries of these sodium polysulfide clusters were optimized to calculate the total energies. The cut-off energy was set to 370 eV for molecular dynamics simulations and the cut-off energy was 450 eV for geometry optimizations aiming to get the accurate energy. A gamma Monkhorst-Pack k-point sampling was used. In this paper, the adsorption energy was defined as: $E(ad) = E(ad/surf) − E(surf) − E(ad)$, where $E(ad/surf)$, $E(surf)$, and $E(ad)$ are the total energies of the adsorbates binding to surface, clean surface and free adsorbate in gas phase, respectively.

## Data availability

The data that support the findings of this work are available from the corresponding author upon reasonable request.

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

## Acknowledgements

This research was supported by the Australian Research Council (ARC) (DE170100928), the Commonwealth of Australia through the Automotive Australia 2020 Cooperative Research Centre (Auto CRC). The authors acknowledge the use of the facilities at the UOW Electron Microscopy Centre funded by ARC grants (LE0882813 and LE0237478) and Dr. Tania Silver for her critical reading.

## Author contributions

B.-W.Z., Y.-X.W., and S.-L.C. conceived and designed the experiments. B.-W.Z. performed all synthetic and characterization experiments. T.S. performed ab initio molecular dynamics simulations. Y.-D.L. performed Raman experiments. L.Z. and W.-H.L. performed the TGA experiments. B.-W.Z., L.W., and Q.-F.G. performed synchrotron X-ray diffraction measurements, and J.Y. performed the ICP measurement. B.-W.Z., Y.-X.W., S.-L.C., H.-K.L., and S.-X.D. analyzed the data and wrote the manuscript. All authors read and approved the final manuscript.
