## [Peer Review File · Nature Communications]

Reviewers' Comments:

Reviewer #1:

Remarks to the Author:

Reviewer's comment for NCOMMS-18-03198

This manuscript reports Co-containing hollow carbon nanospheres (HC) as sulfur host for Na/S (NAS) batteries. Room-temperature (RT) operation of NAS batteries is challenging but quite important, and this study contributes to pushing the research on facile redox reaction of sulfur in NAS batteries at RT. XRD and Raman spectroscopy revealed the reaction mechanism of sulfur. The authors have addressed that the Na₂S₄ moiety is quickly transformed into Na₂S by the added Co component, resulting in the suppression for dissolution of Na₂S₄ into a liquid electrolyte. DFT results support the reaction mechanism which the authors proposed. The revisions based on the following comments are considered before judging for acceptance.

The comments for the manuscript are as follows:

1. The reversible capacity of the NAS cell studied here is still small as shown in Fig. 3. Does the S@Co-HC cell exhibit theoretical capacity of 1672 mAh/g during cycling at a lower current density? The effects of slow charge-discharge process on the suppression of Na₂S₄ dissolution is important and will be discussed in the revised manuscript.
2. The capacity degradation at the initial 10 cycles for the S@Co-HC cell is almost the same as that for the S@HC cell (Fig. 3c), although the initial capacity of the former cell is higher than that of the latter cell. This means the degradation mechanism is similar in the two cells. How about the reason for the capacity degradation? If the suppression of Na₂S₄ with Co-doping to HC affect the initial capacity loss, the profile of the capacity fading should be different.
3. Sulfur at Co-HC is still present as crystalline one because diffraction peaks attributable to crystalline sulfur (S₈) appear in the XRD patterns in Fig. 2b. Amorphous sulfur embedded in carbon spheres would be effective in enhancing cycling performance of NAS batteries. Is it difficult for amorphization of sulfur, which is melt-diffused into pores of HC?
4. Elemental mapping of S@Co-HC as shown in Fig. 1e suggests that sulfur is present not in the carbon sphere but in the carbon layer. How about the reason for no absorption of sulfur in the carbon sphere?
5. In the Co 2p XPS spectrum of S@Co-HC in Fig. 2d, the peaks are quite broad and background is not flat. It is difficult for the peak deconvolution of Co and Co²⁺ in the spectrum.
6. Have the amounts of Co and heat-treatment conditions been optimized for S@Co-HC?
7. Photographs shown in Fig. S14 are difficult to understand. Additional explanation is needed.

Reviewer #2:

Remarks to the Author:

Zhang et al. reported a brand-new research strategy for propelling room-temperature Na-S batteries. Unlike the essential challenge of shuttle effect in Li-S batteries, the priority for room-temperature Na-S batteries is to accelerate the reaction kinetics between Na and S. The work innovatively explored the electrocatalysis effect of atomic Co in S cathode, which successfully enhanced the reaction activity of S and the produced polysulfides. On the other hand, the severe shuttle effect could be simultaneously hindered via the polar-polar interaction due to the surface sulfuration of atomic Co. It is significant that this facile introduction of atomic Co is very effective,

endowing the S cathode with superior Na-storage properties in terms of high accessible capacity, prolonged cycling life, and excellent rate capability. This work is elaborately designed with rational experiments and theoretical calculation. The research results are exciting and well demonstrated in the manuscript. It is a clear and solid work. I think the paper will lead a new research direction and should be of interest to the readers. I recommend accepting it in Nature Communication after minor revision.

1. There are several minor corrections are required. In Figure 1e, the overlay image in the elemental mapping of S@Con-HC should be written in the top right-hand corner. In Figure 3e, the demonstration of "this work" is not clear as there are several samples mentioned in this manuscript.
2. In order to make sure the uniform of the samples, Can the authors provide the SEM images of S@Con-HC?
3. How the authors make sure the Con-HC in S@Con-HC is inactive in the batteries? The cycling performance of the plain matrix is supposed to be displayed in the revised manuscript.
4. New publications in room-temperature Na-S batteries should be cited as well.

Reviewer #3:

Remarks to the Author:

This manuscript reports a cathode material for room-temperature Na-S battery. While the concept is interesting, it brings up a number of questions, which are not clearly addresses. In general, the manuscript is short of the high standard of Nature Communication as it stands now.

(1) the title of the paper reads uncomfortable.

(2) the manuscript lacks a "fig.1" to describe the material synthesis and electrode reaction mechanism in a schematic fashion.

(3) how do S and Co NP enter HC hollow carbon spheres? If HC spheres were indeed spheres, then either the host or the guest must desegregate for the two to merge.

(4) line 125, and line 167-168, "atomic Co" implies these are individual atoms of Co, and they are metallic 0 valence atoms rather than Co ions. How does Co-S fit in this picture of atomic Co?

(5) line 256 on, what are characteristic XRD peaks of longchain polysulfide? aren't they dissolved in electrolyte?

(6) the function of the Co in the cathode material: is it really catalysis? do the authors have any evidence such as CV or LSV to show the catalytic effect? may it be a reaction between Co species and S, rather than catalysis?

Responses to Reviewers:

Reviewer #1 (Remarks to the Author):

Reviewer's comment for NCOMMS-18-03198

This manuscript reports Co-containing hollow carbon nanospheres (HC) as sulfur host for Na/S (NAS) batteries. Room-temperature (RT) operation of NAS batteries is challenging but quite important, and this study contributes to pushing the research on facile redox reaction of sulfur in NAS batteries at RT. XRD and Raman spectroscopy revealed the reaction mechanism of sulfur. The authors have addressed that the Na_2S_4 moiety is quickly transformed into Na_2S by the added Co component, resulting in the suppression for dissolution of Na_2S_4 into a liquid electrolyte. DFT results support the reaction mechanism which the authors proposed. The revisions based on the following comments are considered before judging for acceptance.

The comments for the manuscript are as follows:

1. The reversible capacity of the NAS cell studied here is still small as shown in Fig. 3. Does the $\text{S}@Co_n\text{-HC}$ cell exhibit theoretical capacity of 1672 mAh/g during cycling at a lower current density? The effects of slow charge-discharge process on the suppression of Na_2S_4 dissolution is important and will be discussed in the revised manuscript.

Answer:

Thank you for your helpful comment. In order to understand the impact of low current densities on electrochemical performance of the $\text{S}@Co_n\text{-HC}$ cells, two slow charge/discharge processes have been further tested at 20 and 50 mA g^{-1} , respectively. The electrode could achieve an initial reversible capacity (1613 mAh/g), confirming a high S utilization in $\text{S}@Co_n\text{-HC}$ at low current (20 mA g^{-1}). We have added the corresponding discussion and the effect of slow charge-discharge process on the suppression of Na_2S_4 dissolution in the updated manuscript as follow.

Revised manuscript: “To investigate the effects of slow charge-discharge processes, the S@Co_n-HC cell at low current densities (20 mA g⁻¹ and 50 mA g⁻¹) were carried out, as shown in Fig. S14. It could be clearly seen that the initial reversible capacity of S@Co_n-HC is 1613 mAh g⁻¹ at 20 mA g⁻¹, which is close to the theoretical capacity of S (1672 mAh g⁻¹), retaining reversible capacity of 945 mAh g⁻¹ after 40 cycles. When tested at 50 mA g⁻¹, the S@Co_n-HC delivers an initial reversible capacity of 1360 mAh g⁻¹, maintaining 904 mAh g⁻¹ after 40 cycles. During the slow charge-discharge process at current density of 20 mA g⁻¹, the produced long-chain polysulfides could be further fully sodiated to Na₂S₄. Meanwhile, the atomic Co will effectively alleviate dissolution of Na₂S₄ and electrocatalytically reduce Na₂S₄ into the final product Na₂S. However, the slow charge-discharge process would aggravate the dissolution and shuttle effect of the long-chain polysulfides, leading to fast capacity decay and inferior capacity retention. This phenomenon is in well agreement with the cycling performance, in which this cathode shows the lowest capacity retention (58.5%) at 20 mA g⁻¹. The comparisons at different currents indicate that the slow charge-discharge process is favorable to realize high reversible capacity but severe capacity decay. It is rational to select a current density that would slow enough to exert the capacity of all S active materials and fast enough to alleviate the shuttle effect. By contrast, the current density of 100 mA g⁻¹ shows the most satisfactory performance.”

Figure S14. Discharge/charge curves of S@Co_n-HC at 100 mA g^{-1} (a), 50 mA g^{-1} (c) 20 mA g^{-1} (e). Cycle performance S@Co_n-HC at 100 mA g^{-1} (b), 50 mA g^{-1} (d) 20 mA g^{-1} (f).

2. The capacity degradation at the initial 10 cycles for the S@Co_n-HC cell is almost the same as that for the S@HC cell (Fig. 3c), although the initial capacity of the former cell is higher than that of the latter cell. This means the degradation mechanism is similar in the two cells. How about the reason for the capacity degradation? If the suppression of Na₂S₄ with Co-doping to HC affect the initial capacity loss, the profile of the capacity fading should be different.

Answer:

Thank you for your helpful comment. The discharge/charge profiles from 1st to 10th cycles at 100 mA g⁻¹ of S@Co_n-HC and S@HC cathode materials are shown in below. It is clearly seen that the fade capacity of both two cathode materials take place after the formation of polysulfides. Thus, the main reason for the capacity degradation of S@Co_n-HC and S@HC cathode materials is the same: the surface long-chain polysulfides (generated from the outer S of carbon shell) dissolve in electrolyte, resulting in fast capacity faded during the first 10th cycles. On the other hand, the synchrotron XRD results (Fig. 5c) indicate that the non-conductive Na₂S would accumulate in cathode during the charge/discharge processes, which indicates that the impedance increase in the cell become the main factor for the capacity degradation of S@Co_n-HC and S@HC.

As the S content in S@Co_n-HC (49 %) is higher than that of S@HC (30 %), these two issues are aggravated towards S@Co_n-HC. Significantly, the S@Co_n-HC still outperforms the S@HC with much higher reversible capacity, which is beneficial from the introduction of atomic Co. The schematic illustrations of electrode reaction mechanisms of the S@Co_n-HC and S@HC are shown in Fig. 6d and Fig. 6e. These atomic Co, with surface sulfurization, could effectively prevent the diffusive polysulfides from shuttle effect based on polar-polar interaction. Moreover, the confined polysulfides in the inner carbon shell will be catalyzed into Na₂S by atomic Co. Therefore, the electro-catalytic role of atomic Co will reflect on the plateaus of S@Co_n-HC, which are different from those of S@HC. During the first discharge process, the RT-Na/S@Co_n-HC cell shows two long plateaus that run from 1.68 to 1.04 V, and 1.04 to 0.8 V: the high-voltage plateau corresponds to the solid-liquid transition from S to dissolved long-chain polysulfides; and the low-voltage plateau is attributed to the further sodiation of long-chain polysulfides to short-chain sulfides. By contrast, the two plateaus of S@HC are at 1.82 V and 1.62 V during the initial discharge process. The lower potential plateaus of S@Co_n-HC are likely attributed to the complex bonds between Co and S (Co-S bonds), so that additional energy is

needed to dissociate S from the Co-S bond, resulting in a more negative potential^{1,2}. In addition, for the following cycle, the S@Co_n-HC shows a smoothing plateau from 1.88 to 1.82 V, indicating that the sodiation reaction is fast; however, from 2nd to 10th cycle, the S@HC presents an obvious plateau from 1.86 to 1.80 V, demonstrating that the activity of sodiation reaction is lower than that of S@Co_n-HC. This result also indicates that atomic Co may effectively accelerate the reaction kinetics. We have added this discussion into our revised manuscript.

Revised manuscript: “During the first 10 cycles, there is obvious capacity decay for both of the S@Co_n-HC and S@HC cathodes, which is attributed to the loss of dissolved long-chain polysulfides. The cells show relatively stable cycling but with graduate capacity loss for the subsequent 600 cycles, which mainly originate from the impedance increase in the cells due to the formation of Na₂S. This is consistent with the synchrotron XRD results (Fig. 5), confirming that the non-conductive Na₂S would accumulate in cathode during the charge/discharge processes.”

“The schematic illustrations of electrode reaction mechanisms of the S@Co_n-HC and S@HC are shown in Fig. 6d and Fig. 6e. These atomic Co, with surface sulfurization, could effectively alleviate the diffluent polysulfides dissolution based on polar-polar interaction. Moreover, the confined polysulfides in the inner carbon shell could be fully catalyzed into Na₂S by atomic Co, leading to high S utilization. Therefore, the atomic Co in S@Co_n-HC plays a critical role in achieving sustainable cycling stability and high reversible capacity. By contrast, the intensive “shuttle effect” and incomplete sodiation reactions result in the inferior performance of the S@HC cathode.”

Figure a, b discharge/charge curves of S@Co_n-HC and S@HC at 100 mA g⁻¹.

Figure 6 d, and e, Schematic illustrations of electrode reaction mechanism of S@Co_n-HC and S@HC.

3. Sulfur at Co-HC is still present as crystalline one because diffraction peaks attributable to crystalline sulfur (S₈) appear in the XRD patterns in Fig. 2b. Amorphous sulfur embedded in carbon spheres would be effective in enhancing cycling performance of NAS batteries. Is it difficult for amorphization of sulfur, which is melt-diffused into pores of HC?

Answer:

Thank you for your helpful comment. The amorphous sulfur, which usually exist as a smaller molecule form rather than S₈ (crystalline sulfur), have been demonstrated that could effectivity enhance the cycling performance of NAS batteries³⁻⁵. Xin *et al.* demonstrated that the amorphous sulfur, especially S₂₋₄, should be confined in microporous carbon (~0.5 nm). The synthesis process of high microporous carbon is multi-steps and troublesome. The S loading strategy includes a melt-diffusion process at 155 °C and a complete migration process at 115 °C for 20 hours^{6,7}. Even though amorphous S could show enhanced performance, it is subject to complex preparation procedures and low S loading ratio.

In our samples, we achieved high sulfur loading ratio with the co-existence of amorphous sulfur and crystalline S. Certainly, we also could realize only amorphous sulfur via the complete migration process, but the S loading ratio will be too low (~8

wt%) in S@Co_n-HC. As the TGA results shown in Fig. 3a, there are three states S in S@Co_n-HC. The crystalline S on the carbon layer would sublime at relatively low temperature of ~270 °C, which accounts for S ratio of ~ 33% (crystalline sulfur). Then, small amount of amorphous S, confined in the micropores, would evaporate from 270 to 530 °C with S loss ratio of ~ 8%; the S encapsulated in the hollow space could finally sublime at high temperature from 530 °C, which corresponds to a S portion of ~ 6% S. As for S@HC, the amorphous S is about ~ 4%. We have added this discussion into our revised manuscript.

Revised manuscript: “There are three states of S in the S@Co_n-HC. The crystalline S on the carbon layer would sublime at relatively low temperature of ~270 °C, which accounts for S ratio of ~ 33% (crystalline sulfur). Then, small amount of amorphous S, confined in the micropores¹⁵, would evaporate from 270 to 530 °C with a S loss ratio of ~ 8%; the S encapsulated in the hollow space could finally sublime at high temperature from 530 °C, which corresponds to a S portion of ~ 6% S. The S@HC sample shows the similar TGA curve, indicating S presents the same states as those of the S@Co_n-HC; the amorphous S in S@HC is about ~ 7%.”

Figure 3a. TGA of S@HC and S@Co_n-HC.

4. Elemental mapping of S@Co_n-HC as shown in Fig. 1e suggests that sulfur is present not in the carbon sphere but in the carbon layer. How about the reason for no

absorption of sulfur in the carbon sphere?

Answer:

Thank you for your helpful comment. The S state is determined by our unique synthesis process. Another sample, the S/Co-HC composite heated at 155 °C for 12 h, shows that part of sulfur can aggregate and exist in the carbon sphere and others present in the carbon layer in Fig. S3.

For the final product, the S@Co_n-HC, a thermal treatment at 300 °C for 2 h was further conducted in a sealed quartz ampoule. During the treatment at high temperature (300 °C), the aggregated S in S/Co-HC could sublime. With sufficient thermal energy for S evaporation, most of the S molecules diffuse into the C shells, and are adsorbed by atomic Co as shown in Fig. 2. Hence, for S@Co_n-HC, there is only small amount of sulfur inside of the carbon sphere, as displayed in TGA curves. In order to make it clear, we have added schematic illustrations of this process and corresponding demonstrations into the revised manuscript.

Revised manuscript: “The synthetic process of the S@Co_n-HC is illustrated in Fig. 1. The successful encapsulation of Co nanoparticles (NPs, ~3 nm) and S is attributed to the microporous and hollow structure of carbon spheres. Initially, the CoCl₂ solution can be immersed and impregnated into the HC spheres, which was reduced to Co NPs uniformly decorated on the C shells (~5 nm) of HC nanospheres (Co-HC) by controlled thermal treatment method (Fig. S1 and Fig. S2, Supporting Information). The interaction between Co and S undergoes two stages along with increasing temperature. Firstly, the melted S was loaded into the Co-HC by capillarity effect via a facile melt-diffusion strategy at 155 °C for 12 h (with the product denoted as S/Co-HC). It is clear that part of S agglomerate in the hollow space of carbon spheres and others are dispersed in the carbon shells for the S/Co-HC as shown in atomic resolution high-angle annular dark field (HAADF) scanning transmission electron microscopy (STEM) images (Fig. S3). Subsequently, the S/Co-HC was heat-treated at 300 °C in a sealed quartz ampoule, which interestingly leads to the disappearance of Co nanoparticles and S agglomeration. During this process, S begins to sublime. The concentration gradient results in S diffusion from the inside of the nanospheres to

the surface. With sufficient thermal energy for S evaporation, most of the S molecules diffuse into the C shells, which would drive the Co nanoparticles to be re-dispersed into the carbon shells as well. Thus, atomic Co, including Co single atoms and clusters, migrates into the C shells of each HC nanosphere by taking advantage of the diffusion of inner S molecules. Finally, a novel S nanocomposite with S embedded into atomic Co decorated hollow carbon ($S@Co_n\text{-HC}$) could be achieved.”

Figure 1. Schematic illustration of the synthesis of the $S@atomic\ Co$ decorated hollow carbon nanocomposite ($S@Co_n\text{-HC}$).

Figure S3. a, HADF-STEM images of S/Co-HC. b-e, elemental mapping of S/Co-HC.

5. In the Co 2p XPS spectrum of S@Co-HC in Fig. 2d, the peaks are quite broad and background is not flat. It is difficult for the peak deconvolution of Co and Co²⁺ in the spectrum.

Answer:

Thank you for your helpful comment. We have retested the XPS, as shown in Fig. 3c and 3d. The Co 2p XPS of S@Co_n-HC indicated that the atomic Co have been deconvoluted into Co⁰ (778.70 eV) and Co²⁺ (781.60 eV). The binding energy of the Co⁰ 2p_{3/2} XPS peak is 778.20 eV in pure Co; while the binding energy in S@Co_n-HC is 778.70 eV, which indicates the formation of Co-S bonds in S@Co_n-HC, demonstrating that the “anchor” S will affect the electronic structure of Co. This result has been added into our revised manuscript.

Revised manuscript: “The Co 2p XPS of S@Co_n-HC in Fig. 3d indicated that the atomic Co have been deconvoluted into Co⁰ (778.70 eV) and Co²⁺ (781.60 eV). The Co²⁺ (781.60 eV) in S@Co_n-HC could be attributed to single Co atoms anchored on S dispersed hollow carbon³⁴, probably formation of Co-S bond. While, except for single Co atoms, S@Co_n-HC exists Co clusters as shown in Fig. 2 and Fig. S4. Because of these Co clusters existing, Co 2p XPS of S@Co_n-HC will present the Co⁰ state. The binding energy of the Co⁰ 2p_{3/2} XPS peak in S@Co_n-HC is 778.70 eV, which right shift 0.5 eV compared with that of pure Co (778.20 eV); this right shift binding energy indicates the formation of Co-S bonds between Co clusters and S in S@Co_n-HC.”

Figure 3. **c**, S 2p XPS spectra for S@HC and S@Co_n-HC respectively. **d**, Co 2p XPS spectra for S@Co_n-HC.

6. Have the amounts of Co and heat-treatment conditions been optimized for S@Co-HC?

Answer:

Thank you for your helpful comment. For the S@Co_n-HC, the ICP-OES result demonstrates that the Co content is 7.06 wt. %, which is prepared from 10 wt. % of CoCl₂. For optimization, 5% and 20% of CoCl₂ are applied via the same synthesis procedures. ICP-OES results show that the weight content of Co in these two samples are 2.33% and 15.02%, named as S@Co_n-HC-2 and S@Co_{NP}-HC-15 respectively. The HAADF-STEM with elemental mapping of S@Co_n-HC-2 in Fig. S7 demonstrate the similar morphologies and components to that of the S@Co_n-HC. Atomic Co is well confined in the carbon shells, and sulfur is also well dispersed on the carbon shell; meanwhile, the Co elemental mapping in Fig. S8 also indicates that the content of Co is lower than S@Co_n-HC. However, the HAADF-STEM results of S@Co_{NP}-HC-15 in Fig. S8 indicate that Co nanoparticles are formed at this high Co content. The TGA results of these different Co content cathode materials in Fig. S9 indicate that the S contents in S@Co_n-HC-2, S@Co_n-HC, and S@Co_{NP}-HC-15 are ~ 36 %, 47 %, and 47 %, respectively. The low S loading ratio of S@Co_n-HC-2 (36%) indicates that high atomic Co content in HC is favourable to capture S and enhance S loading amount; it is noteworthy that the sulfur loading ratio of S@Co_{NP}-HC-15 equal with that of S@Co_n-HC, demonstrating that only simply increasing the Co content can not enhance the sulfur loading.

To investigate the impact of Co contents on electrochemical performance, their CVs and discharge/charge profiles of the 1st, 2nd, 10th, 50th, 100th and 200th cycles at 100 mA g⁻¹ of S@Co_n-HC-2, S@Co_n-HC and S@Co_{NP}-HC-15 cathode materials are shown in Fig. S15. The CV of S@Co_n-HC-2 has three peaks at 1.97, 1.49, and 1.09 V during cathodic cycling and three peaks at 1.55, 1.75, and 2.22 V in the anodic scan, indicating that S is reduced in an orderly manner to Na₂S_x, Na₂S₄, and Na₂S. It is interesting that CV peaks of S@Co_{NP}-HC-15 are almost the same, indicating that they may go through the same mechanism. The RT-Na/S@Co_n-HC-2 cell shows three long

plateaus from 1.91 to 1.50 V, 1.50 to 1.10 V and 1.10 to 0.8 V during the initial discharge process, corresponding to its CV results. Meanwhile, the S@Co_{NP}-HC-15 presents four plateaus from 2.28 to 2.20 V, 1.81 to 1.77 V, 1.71 to 1.66 V, and 1.66 to 0.8 V during the initial discharge process. The lower potential plateaus of S@Co_n-HC-2 in the initial cycle may be attributed to that the atomic Co has a stronger ability to immobilize S than Co nanoparticles, even the Co content of S@Co_n-HC-2 (2.33%) is quite lower than that of S@Co_{NP}-HC-15 (15.02%). The initial reversible capacity of S@Co_n-HC-2 and S@Co_{NP}-HC-15 are 976 and 780 mAh g⁻¹, respectively; meanwhile they retain the reversible capacity of 383 and 438 mAh g⁻¹ after 200 cycles. Significantly, the S@Co_n-HC processes the best performance among these cathode materials, as shown in Fig. S15.

As shown in supporting information, we have optimized the heat-treatment conditions, the sample with 10% CoCl₂ precursor in the sealed quartz ampoule was processed by thermal treatment at 155 °C, 300 °C and 400 °C for 2 h, respectively, which lead to the formation of three samples, including S/Co-HC, S@Co_n-HC, and S@CoS₂-HC. The HAADF-STEM results of S/Co-HC, S@Co_n-HC and S@CoS₂-HC are shown in Fig. S5, Fig. 2 and Fig. S6, respectively. It could be clearly seen that for 155 °C treatment (S/Co-HC), part of sulfur will aggregate in the carbon sphere and others will present in the carbon layer; after thermal treatment at 300 °C (S@Co_n-HC), atomic Co is embedded in the carbon shells, as shown in Fig. 2. While at 400 °C, S will react with Co to form of CoS₂, in Fig. S6. TGA results in Fig. S10 shows that the S contents in S/Co-HC, S@Co_n-HC, and S@CoS₂-HC is ~ 48 %, 47 %, and 30 %, respectively. The discharge/charge profiles and cycle performance of S/Co-H, S@Co_n-HC and S@CoS₂-HC at 100 mA g⁻¹ are shown in Fig. S13. The S@Co_n-HC delivers an initial reversible capacity of 1081 mAh g⁻¹, retaining excellent reversible capacity of 508 mAh g⁻¹ after 600 cycles. To highlight the role of atomic Co, the cycling stability of S/Co-HC and S@CoS₂-HC are shown in Fig. S16. It is noteworthy that the S/Co-HC displays fast capacity degradation, which shows the initial reversible capacity of 1018/617 mAh g⁻¹, but after 100 cycles, it is only 64/62 mAh g⁻¹. Additionally, the first cycle reversible capacity of S@CoS₂-HC is 610/1415

respectively; after 500 cycles, it is only 173 mAh g⁻¹. These results demonstrate that the atomic Co possesses stronger electrocatalytic capability than Co nanoparticles and CoS₂ nanoparticles. We have added this discuss into revised manuscript and supporting information.

Revised manuscript: “Meanwhile, the Co loading ratios (5% and 20% of CoCl₂) also have been optimized for S@Co-HC as shown in Fig. S7 and Fig. S8.”

Supporting information: “For the S@Co_n-HC, the ICP-OES result demonstrates that the Co content is 7.06 wt. %, which is prepared form 10 wt. % of CoCl₂. For optimization, 5% and 20% of CoCl₂ are applied via the same synthesis procedures. ICP-OES results show that the weight content of Co in these two sample are 2.33% and 15.02%, named as S@Co_n-HC-2 and S@Co_{NP}-HC-15 respectively. The HAADF-STEM with elemental mapping of S@Co_n-HC-2 in Fig. S7 demonstrate the similar morphologies and components to that of the S@Co_n-HC. Atomic Co is well confined in the carbon shells, and sulfur is also well dispersed on the carbon shell; meanwhile, the Co elemental mapping in Fig. S8 also indicates that the content of Co is lower than S@Co_n-HC. However, the HAADF-STEM results of S@Co_{NP}-HC-15 in Fig. S8 indicate that Co nanoparticles are formed at this high Co content. The TGA results of these different Co content cathode materials in Fig. S9 indicate that the S contents in S@Co_n-HC-2, S@Co_n-HC, and S@Co_{NP}-HC-15 are ~ 36 %, 47 %, and 47 %, respectively. The low S loading ratio of S@Co_n-HC-2 (36%) indicates that high atomic Co content in HC is favourable to capture S and enhance S loading amount; it is noteworthy that the sulfur loading ratio of S@Co_{NP}-HC-15 equal with that of S@Co_n-HC, demonstrating that only simply increasing the Co content can not enhance the sulfur loading.”

“To investigate the impact of Co contents on electrochemical performance, their CVs and discharge/charge profiles of the 1st, 2nd, 10th, 50th, 100th and 200th cycles at 100 mA g⁻¹ of S@Co_n-HC-2, S@Co_n-HC and S@Co_{NP}-HC-15 cathode materials are shown in Fig. S15. The CV of S@Co_n-HC-2 has three peaks at 1.97, 1.49, and 1.09 V during cathodic cycling and three peaks at 1.55, 1.75, and 2.22 V in the anodic scan, indicating that S is reduced in an orderly manner to Na₂S_x, Na₂S₄, and Na₂S. It is

interesting that CV peaks of S@Co_{NP}-HC-15 are almost the same, indicating that they may go through the same mechanism. The RT-Na/S@Co_n-HC-2 cell shows three long plateaus from 1.91 to 1.50 V, 1.50 to 1.10 V and 1.10 to 0.8 V during the initial discharge process, corresponding to its CV results. Meanwhile, the S@Co_{NP}-HC-15 presents four plateaus from 2.28 to 2.20 V, 1.81 to 1.77 V, 1.71 to 1.66 V, and 1.66 to 0.8 V during the initial discharge process. The lower potential plateaus of S@Co_n-HC-2 in the initial cycle may be attributed to that the atomic Co has a stronger ability to immobilize S than Co nanoparticles, even the Co context of S@Co_n-HC-2 (2.33%) is quite lower than that of S@Co_{NP}-HC-15 (15.02%). The initial reversible capacity of S@Co_n-HC-2 and S@Co_{NP}-HC-15 are 976 and 780 mAh g⁻¹, respectively; meanwhile they retain the reversible capacity of 383 and 438 mAh g⁻¹ after 200 cycles. Significantly, the S@Co_n-HC processes the best performance among these cathode materials, as shown in Fig. S15.”

Figure S7. a, HADDF-STEM images of S@Co_x-HC. b-e, elemental mapping of S@Co_x-HC.

Figure S8. a, HADDF-STEM images of S@Co_{NP}-HC. b-e, elemental mapping of S@Co_{NP}-HC.

Figure S9. TGA of S@Co_n-HC-2, S@Co_n-HC and S@Co_{NP}-HC.

Figure S15. a and b, Cyclic voltammograms of S@Co_n-HC-2 and S@Co_{NP}-HC-15 within the voltage window of 0.8 - 2.8 V at a scan rate of 0.1 mV s⁻¹. **c and d,** Discharge/charge curves of S@Co_n-HC-2 and S@Co_{NP}-HC-15 at 100 mA g⁻¹. **e,** Cycling performance of S@Co_n-HC-2, S@Co_{NP}-HC-15, S@Co_n-HC and S@HC at 100 mA g⁻¹.

Figure S6. a-b, HAADF-STEM images of S@CoS₂-HC. c, line-profile analysis of the indicated of line in a. d-h elemental mapping of S@CoS₂-HC.

Figure S10. Thermogravimetry (TGA) curves of S/Co-HC, S@Co_n-HC, and S@CoS₂-HC.

Figure S13. Discharge/charge voltage profiles of RT-Na/S cell of a, S/Co -HC, b, S@CoS₂-HC at selected cycles at 100 mA g⁻¹.

Figure S16. Cycling performance at 100 mA g⁻¹ of S/Co-HC, S@Co_n-HC and S@CoS₂-HC.

7. Photographs shown in Fig. S14 are difficult to understand. Additional explanation is needed.

Answer:

Thank you for your helpful comment. According to the comment, we have extended the explanation, as below:

Revised manuscript: “Furthermore, the polysulfide dissolution behaviours of S@Co_n-HC and S@HC electrodes using transparent glass cells are compared in Fig.

S19. The cell with S@Co_n-HC maintained colourless during the 10 h discharge process, which implies the alleviation of the polysulfide dissolution, suggesting that atomic Co could kinetically catalyze the polysulfide reduction into Na₂S instead of dissolution into electrolyte. However, the yellow polysulfide in the surface of S@HC electrode was observed upon discharged for 5 h; when upon for 10 h sodiation process, it could be clearly seen that yellow polysulfide was dissolved in the cell. This color change of S@HC indicates that the polysulfide dissolution into electrolyte, i.e. shuttle effect, which could lead to loss of active materials.”

References

- 1 Wang, J. *et al.* Sulfur Composite Cathode Materials for Rechargeable Lithium Batteries. *Advanced Functional Materials* **13**, 487-492, doi:10.1002/adfm.200304284 (2003).
- 2 Zhang, B., Qin, X., Li, G. R. & Gao, X. P. Enhancement of long stability of sulfur cathode by encapsulating sulfur into micropores of carbon spheres. *Energy & Environmental Science* **3**, 1531-1537, doi:10.1039/C002639E (2010).
- 3 Qiang, Z. *et al.* Ultra-long cycle life, low-cost room temperature sodium-sulfur batteries enabled by highly doped (N,S) nanoporous carbons. *Nano Energy* **32**, 59-66, doi:https://doi.org/10.1016/j.nanoen.2016.12.018 (2017).
- 4 Zheng, S. *et al.* Nano-Copper-Assisted Immobilization of Sulfur in High-Surface-Area Mesoporous Carbon Cathodes for Room Temperature Na-S Batteries. *Advanced Energy Materials* **4**, 1400226, doi:10.1002/aenm.201400226 (2014).
- 5 Hwang, T. H., Jung, D. S., Kim, J. S., Kim, B. G. & Choi, J. W. One-dimensional carbon-sulfur composite fibers for Na-S rechargeable batteries operating at room temperature. *Nano Lett* **13**, 4532-4538, doi:10.1021/nl402513x (2013).
- 6 Xin, S., Yin, Y. X., Guo, Y. G. & Wan, L. J. A high-energy room-temperature sodium-sulfur battery. *Adv Mater* **26**, 1261-1265,

doi:10.1002/adma.201304126 (2014).

- 7 Xin, S. *et al.* Smaller sulfur molecules promise better lithium-sulfur batteries. *J Am Chem Soc* **134**, 18510-18513, doi:10.1021/ja308170k (2012).
- 8 Wang, Y. X. *et al.* Achieving High-Performance Room-Temperature Sodium-Sulfur Batteries With S@Interconnected Mesoporous Carbon Hollow Nanospheres. *J Am Chem Soc* **138**, 16576-16579, doi:10.1021/jacs.6b08685 (2016).

Reviewer #2 (Remarks to the Author):

Zhang et al. reported a brand-new research strategy for propelling room-temperature Na-S batteries. Unlike the essential challenge of shuttle effect in Li-S batteries, the priority for room-temperature Na-S batteries is to accelerate the reaction kinetics between Na and S. The work innovatively explored the electrocatalysis effect of atomic Co in S cathode, which successfully enhanced the reaction activity of S and the produced polysulfides. On the other hand, the sever shuttle effect could be simultaneously hindered via the polar-polar interaction due to the surface sulfuration of atomic Co. It is significant that this facile introduction of atomic Co is very effective, endowing the S cathode with superior Na-storage properties in terms of high accessible capacity, prolonged cycling life, and excellent rate capability. This work is elaborately designed with rational experiments and theoretical calculation. The research results are exciting and well demonstrated in the manuscript.

It is a clear and solid work. I think the paper will lead a new research direction and should be of interest to the readers. I recommend accepting it in Nature Communication after minor revision.

1. There are several minor corrections are required. In Figure 1e, the overlay image in the elemental mapping of S@Co_n-HC should be written in the top right-hand corner. In Figure 3e, the demonstration of “this work” is not clear as there are several samples mentioned in this manuscript.

Answer:

Thank you for your helpful comment. We have written the “overlay” in the elemental mapping of S@Co_n-HC in Figure 2h.

In Figure 4e, “this work” has revised into “this work S@Co_n-HC”.

Revised manuscript:

Figure 2 | HADDF-STEM images. a, b, c TEM and HADDF-STEM images of S@Co_n-HC. d, line-profile analysis from the indicated of b. e-h, elemental mapping of S@Co_n-HC.

Figure 4 | RT-Na/S batteries best. a, b discharge/charge curves of S@Co_n-HC and S@HC at 100 mA g⁻¹. c, d, cycle performance and rate performance for S@Co_n-HC and S@Co. e, Comparison of the rate and cycling (inset) capabilities of the previously reported RT-Na/S batteries with our work.

2. In order to make sure the uniform of the samples, Can the authors provide the SEM images of S@Co_n-HC?

Answer:

Thank you for your helpful comment. We have provided the SEM image and TEM images of S@Co_n-HC as shown in Fig. S4. SEM and TEM images demonstrate that the hollow carbon is uniform, and there are no big Co nanoparticles. And we have added these images into Supporting Information.

Revised manuscript: “SEM and TEM images demonstrate that the hollow carbon is uniform, and there are no big Co nanoparticles” .

Supporting information:

Figure S4. a, SEM, b, TEM and c, HADDF-STEM image of S@Co_n-HC. d, Histogram showing S@Co_n-HC distribution based on a count of 200 clusters in the sample areas.

3. How the authors make sure the Co_n-HC in S@Co_n-HC is inactive in the batteries? The cycling performance of the plain matrix is supposed to be displayed in the revised manuscript.

Answer:

Thank you for your careful reading. In order to make sure the Co_n-HC in S@Co_n-HC

is inactive in the RT/Na-S batteries, we have studied the electrochemical performance of the plain matrix, in which S is removed from S@Co_n-HC to form Co_n-HC. The removing S experimental process is following:

30 mg S@Co_n-HC was dispersed in 30 mL CS₂ solution. The mixture was ultrasonicated for 2 h, followed by washing by ethanol, acetone and DI water with several times, respectively. Finally, the obtained black powder was heated overnight in an oven at 80 °C for 12 h.

The XRD results of Co_n-HC and S@Co_n-HC are shown in Fig. S20. It could be clearly seen that Co_n-HC doesn't show any S characteristic peak, indicating that S has been completely removed. To investigate the electrochemical performance of Co_n-HC, the discharge/charge profiles and cycle performance of Co_n-HC cathode materials are shown in Fig. S21. The Co_n-HC displays an initial reversible capacity of 70 mAh g⁻¹, which is extremely lower than that of S@Co_n-HC (1081 mAh g⁻¹). Meanwhile, the Co_n-HC only maintains a reversible capacity of 40.1 mAh g⁻¹ after 200 cycles. This low capacity could be ignored when compared with that of S@Co_n-HC, which also possesses an excellent capacity of 508 mAh g⁻¹ after 600 cycles. Therefore, it could be clear that Co_n-HC in S@Co_n-HC is inactive in the batteries. We have added this discussion into our revised manuscript.

Revised manuscript: “In order to guarantee the capacity reliability of the S@Co_n-HC cathode, the capacity contribution of S host, the Co_n-HC, was evaluated as well. The Co_n-HC was fabricated from the S@Co_n-HC sample by solving the loaded S away by CS₂ solvent. The XRD results of Co_n-HC and S@Co_n-HC are shown in Fig. S20. It could be clearly seen that Co_n-HC does not show any S characteristic peak, indicating that S has been completely removed. The discharge/charge profiles and cycle performance of Co_n-HC are shown in Fig. S21a, which displays an very low initial reversible capacity of 70 mAh g⁻¹, only retaining a reversible capacity of 40.1 mAh g⁻¹ after 200 cycles. By contrast, Fig. S21b clearly displays that the capacity contribution of Co_n-HC in the S@Co_n-HC cathode could be negligible.”

Supporting information:

Figure S20. XRD of $\text{Co}_n\text{-HC}$ and $\text{S@Co}_n\text{-HC}$.

Figure S21. **a**, discharge/charge curves and **b**, cycle performance of $\text{Co}_n\text{-HC}$ and $\text{S@Co}_n\text{-HC}$ at 100 mA g^{-1} .

4. New publications in room-temperature Na-S batteries should be cited as well.

Answer:

Thank you for your helpful comment. We have cited the new publication on room-temperature Na-S batteries, and their capacities have been compared with our work in our revised manuscript, as shown in Fig. 4e.

Revised manuscript:

Figure 4e. Comparison of the rate and cycling (inset) capabilities of the previously reported RT-Na/S batteries with our work.

43 Qiang, Z. *et al.* Ultra-long cycle life, low-cost room temperature sodium-sulfur batteries enabled by highly doped (N,S) nanoporous carbons. *Nano Energy* **32**, 59-66, doi:<https://doi.org/10.1016/j.nanoen.2016.12.018> (2017).

44 Yao, Y. *et al.* Binding S_{0.6}Se_{0.4} in 1D Carbon Nanofiber with C-S Bonding for High-Performance Flexible Li-S Batteries and Na-S Batteries. *Small* **13**, 1603513, doi:[10.1002/sml.201603513](https://doi.org/10.1002/sml.201603513) (2017).

45 Lu, Q. *et al.* Freestanding carbon fiber cloth/sulfur composites for flexible room-temperature sodium-sulfur batteries. *Energy Storage Materials* **8**, 77-84, doi:<https://doi.org/10.1016/j.ensm.2017.05.001> (2017).

46 Lee, D.-J. *et al.* Alternative materials for sodium ion-sulphur batteries. *Journal of Materials Chemistry A* **1**, 5256, doi:[10.1039/c3ta10241f](https://doi.org/10.1039/c3ta10241f) (2013).

Reviewer #3 (Remarks to the Author):

This manuscript reports a cathode material for room-temperature Na-S battery. While the concept is interesting, it brings up a number of questions, which are not clearly addresses. In general, the manuscript is short of the high standard of Nature Communication as it stands now.

(1) The title of the paper reads uncomfortable.

Answer: Thank you very much for your helpful comment. We have revised the title into “Atomic cobalt as an efficient electrocatalyst in sulfur cathodes for superior room-temperature sodium-sulfur batteries”

(2) The manuscript lacks a "fig.1" to describe the material synthesis and electrode reaction mechanism in a schematic fashion.

Answer:

Thank you for your careful comment. We have added the schemes of the materials synthesis (Fig. 1) and electrode reaction mechanisms (Fig. 6e and 6e) in our revised manuscript as below.

Revised manuscript:

Figure 1. Schematic illustration of the synthesis of the S@atomic Co decorated hollow carbon nanocomposite (S@Co_n-HC).

The synthetic process of the S@Co_n-HC is illustrated in Fig. 1. The successful encapsulation of Co nanoparticles (NPs, ~3 nm) and S is attributed to the microporous and hollow structure of carbon spheres. Initially, the CoCl₂ solution can be immersed and impregnated into the HC spheres, which was reduced to Co NPs uniformly decorated on the C shells (~5 nm) of HC nanospheres (Co-HC) by controlled thermal treatment method (Fig. S1 and Fig. S2, Supporting Information). The interaction between Co and S undergoes two stages along with increasing temperature. Firstly, the melted S was loaded into the Co-HC by capillarity effect via a facile melt-diffusion strategy at 155 °C for 12 h (with the product denoted as S/Co-HC). It is clear that part of S agglomerate in the hollow space of carbon spheres and others are dispersed in the carbon shells for the S/Co-HC as shown in atomic resolution high-angle annular dark field (HAADF) scanning transmission electron microscopy (STEM) images (Fig. S3). Subsequently, the S/Co-HC was heat-treated at 300 °C in a sealed quartz ampoule, which interestingly leads to the disappearance of Co nanoparticles and S agglomeration. During this process, S begins to sublime. The concentration gradient results in S diffusion from the inside of the nanospheres to the surface. With sufficient thermal energy for S evaporation, most of the S molecules diffuse into the C shells, which would drive the Co nanoparticles to be re-dispersed into the carbon shells as well. Thus, atomic Co, including Co single atoms and clusters, migrates into the C shells of each HC nanosphere by taking advantage of the diffusion of inner S molecules. Finally, a novel S nanocomposite with S embedded into atomic Co decorated hollow carbon (S@Co_n-HC) could be achieved.

“The schematic illustrations of electrode reaction mechanisms of the S@Co_n-HC and S@HC are shown in Fig. 6d and Fig. 6e. These atomic Co, with surface sulfurization, could effectively alleviate the diffluent polysulfides dissolution based on polar-polar interaction. Moreover, the confined polysulfides in the inner carbon shell could be fully catalyzed into Na₂S by atomic Co, leading to high S utilization. Therefore, the atomic Co in S@Co_n-HC plays a critical role in achieving sustainable cycling stability and high reversible capacity. By contrast, the intensive “shuttle effect” and

incomplete sodiation reactions result in the inferior performance of the S@HC cathode.”

Figure 6 | d, and e, Schematic illustrations of electrode reaction mechanism of S@Co_n-HC and S@HC.

(3) How do S and Co NP enter HC hollow carbon spheres? If HC spheres were indeed spheres, then either the host or the guest must desegregate for the two to merge.

Answer:

Thank you for your helpful comment. The encapsulation of S and Co NPs is attributed to the microporous and hollow structure of carbon spheres. The Co NPs source, CoCl₂ solution, can be immersed and impregnated into the HC spheres. We used the controlled thermal treatment method, to synthesize Co NPs supported into the HC, which has been proved as an effective method for nanoparticles loaded on carbon materials¹⁻³. The S encapsulation is achieved by a conventional melt-diffusion strategy at 155 °C for 12h, in which melted S can be loaded in HC by capillarity effect. Accordingly, the synthesis process has been discussed in details as illustrated in Fig.1. Some changes have been added in the revised manuscript as well.

Revised manuscript: “The synthetic process of the S@Co_n-HC is illustrated in Fig. 1. The successful encapsulation of Co nanoparticles (NPs, ~3 nm) and S is attributed to the microporous and hollow structure of carbon spheres. Initially, the CoCl₂ solution can be immersed and impregnated into the HC spheres, which was reduced to Co NPs uniformly decorated on the C shells (~5 nm) of HC nanospheres (Co-HC) by controlled thermal treatment method (Fig. S1 and Fig. S2, Supporting Information).”

Figure S1. a-c, TEM images of HC. d, Histogram showing wall thickness distribution based on a count of 200 wall thick in the sample areas.

Figure S2. a-c, Low and high magnification TEM images of Co-HC. d, Histogram showing Co nanoparticles distribution based on a count of 200 Co nanoparticles in the sample areas.

“Firstly, the melted S was loaded into the Co-HC by capillarity effect via a facile melt-diffusion strategy at 155 °C for 12 h (with the product denoted as S/Co-HC). It is clear that part of S agglomerate in the hollow space of carbon spheres and others are dispersed in the carbon shells for the S/Co-HC as shown in atomic resolution high-angle annular dark field (HAADF) scanning transmission electron microscopy (STEM) images (Fig. S3).”

Figure S3. a, HADDF-STEM images of S/Co-HC. **b-e,** elemental mapping of S/Co-HC.

(4) line 125, and line 167-168, "atomic Co" implies these are individual atoms of Co, and they are metallic 0 valence atoms rather than Co ions. How does Co-S fit in this picture of atomic Co?

Answer:

Thank you for your helpful comment. The Co 2p XPS of S@Co_n-HC in Fig. 3d indicated that the atomic Co have been deconvoluted into Co⁰ (778.70 eV) and Co²⁺ (781.60 eV). It is well-known that single metal atoms are active and unsaturated coordination environment, which could achieve the maximize atom efficiency^{4,5}; nevertheless, this characteristic also result in that single metal atoms are instability^{6,7}. In order to obtain stable single metal atoms in carbon materials, heteroatoms (like N atoms) are usually introduced to serves as an "anchor" to stabilize these single metal atoms^{6,8-10}. Therefore, these single metal atoms are likely to be in a charge state, such as single Ni⁺ atoms anchored on the N-doped graphene matrix¹¹ and single Co (the Co ion is between Co⁰ and Co²⁺) atoms anchored on hollow N-doped porous carbon spheres⁸. Therefore, the Co²⁺ (781.60 eV) in S@Co_n-HC could be attributed single Co atoms anchored on S dispersed hollow carbon¹⁰, probably formation of Co-S bond. While, except for single Co atoms, S@Co_n-HC exists Co clusters as shown in Fig. 2

and Fig. S4. The average size of the atomic Co is calculated to be 0.4 ± 0.2 nm from 200 single atoms and clusters in Fig. S4. Because of these Co clusters existing, Co 2p XPS of S@Co_n-HC will present the Co⁰ state. The binding energy of the Co⁰ 2p_{3/2} XPS peak in S@Co_n-HC is 778.70 eV, which right shift 0.5 eV compared with that of pure Co (778.20 eV); this right shift binding energy indicates the formation of Co-S bonds between Co clusters and S in S@Co_n-HC. We have added this discussion into our revised manuscript.

Revised manuscript: “The Co 2p XPS of S@Co_n-HC in Fig. 3d indicated that the atomic Co have been deconvoluted into Co⁰ (778.70 eV) and Co²⁺ (781.60 eV). The Co²⁺ (781.60 eV) in S@Co_n-HC could be attributed to single Co atoms anchored on S dispersed hollow carbon³⁴, probably formation of Co-S bond. While, except for single Co atoms, S@Co_n-HC exists Co clusters as shown in Fig. 2 and Fig. S4. Because of these Co clusters existing, Co 2p XPS of S@Co_n-HC will present the Co⁰ state. The binding energy of the Co⁰ 2p_{3/2} XPS peak in S@Co_n-HC is 778.70 eV, which right shift 0.5 eV compared with that of pure Co (778.20 eV); this right shift binding energy indicates the formation of Co-S bonds between Co clusters and S in S@Co_n-HC.”

Figure 3c, S 2p XPS spectra for S@HC and S@Co_n-HC respectively. **d**, Co 2p XPS spectra for S@Co_n-HC.

(5) line 256 on, what are characteristic XRD peaks of longchain polysulfide? aren't

they dissolved in electrolyte?

Answer:

Thank you for your helpful comment. The *in situ* synchrotron XRD patterns and enlarged region of the 23° of the RT-Na/S@Co_n-HC cells are shown in Fig. 5c. The peak at 23.01° is indexed to the (240) planes of S (JCPDF no. 71-0569); discharged to 1.8V, the peak at 23.01° became broad, and a new peak (22.97°) evolves around the original peak (23.01°), as shown below in Fig. A1. However, the XRD JCPDS cards do not have the information of long-chain polysulfides. According to the previous work on *in situ* synchrotron XRD results of S@interconnected mesoporous carbon hollow nanospheres¹², the similar peak at 22.97° could be likely attributed to long-chain polysulfides.

We applied the in-situ synchrotron XRD to monitor the compositional and structural changes based on a Na-S cell instead of a S cathode. Even though the polysulfides could dissolve into electrolyte, we can still collect the signals of polysulfides since all the reactions occur inside of a coin-cell device.

Figure 5c, *in situ* synchrotron XRD patterns of the RT-Na/S@Co_n-HC cells (left) with the initial galvanostatic charge/discharge curve (middle) at 500 mA g^{-1} , contour plot of XRD spectra with selected theta (right) at 100 mA g^{-1} .

Figure A1. A larger image at round 22.5° to 24.0° for the first 9 recorded line of *in situ* synchrotron XRD.

(6) The function of the Co in the cathode material: is it really catalysis? do the authors have any evidence such as CV or LSV to show the catalytic effect? may it be a reaction between Co species and S, rather than catalysis?

Answer:

Thank you for your helpful comment. The CVs of S@Co_n-HC and S@HC cathode materials is shown in Fig. 5a and Fig. S18c. The RT-Na/S@Co_n-HC cell shows two prominent peaks at around 1.68 and 1.04 V during the first cathodic scan. The RT-Na/S@HC cell presents four peaks at around 2.20, 1.78, 1.66, and 1.01 V during the initial cathodic scan. These two peaks of S@Co_n-HC indicate that the reaction rate between S and Na is quick, which results in that no obvious peaks of Na₂S₄ and short-chain polysulfides (Na₂S_y, 1 < y ≤ 3). The four peaks of the S@HC demonstrate that the kinetic of sodiation is sluggish, compared with S@Co_n-HC. This result indicates that atomic Co is likely to catalytic reduced polysulfide into the final product Na₂S.

In a Na-storage battery, no any reaction between Co and S could take place. In order to make sure S host, the Co_n-HC, in S@Co_n-HC is inactive in the RT/Na-S batteries,

we have studied the electrochemical performance of the plain matrix, in which S is removed from S@Co_n-HC to form Co_n-HC. The removing S experimental process is following:

30 mg S@Co_n-HC was dispersed in 30 mL CS₂ solution. The mixture was ultrasonicated for 2 h, followed by washing by ethanol, acetone and DI water with several times, respectively. Finally, the obtained black powder was heated overnight in an oven at 80 °C for 12 h.

The XRD results of Co_n-HC and S@Co_n-HC are shown in Fig. S20. It could be clearly seen that Co_n-HC doesn't show any S characteristic peak, indicating that S has been completely removed. To investigate the electrochemical performance of Co_n-HC, the discharge/charge profiles and cycle performance of Co_n-HC cathode materials are shown in Fig. S21. The Co_n-HC displays an initial reversible capacity of 70 mAh g⁻¹, which is extremely lower than that of S@Co_n-HC (1081 mAh g⁻¹). Meanwhile, the Co_n-HC only maintains a reversible capacity of 40.1 mAh g⁻¹ after 200 cycles. This low capacity could be ignored when compared with that of S@Co_n-HC, which also possesses an excellent capacity of 508 mAh g⁻¹ after 600 cycles. Therefore, it could be clear that Co_n-HC in S@Co_n-HC is inactive in the batteries. On the other hand, compared with S@HC, these atomic Co in S@Co_n-HC could effectively enhance RT/Na-S@Co_n-HC batteries' capacity; therefore, these atomic Co plays the "catalysts" role in the cathode material. The function of the atomic Co in the cathode material has been summarized in Fig. 6d and 6e. These atomic Co, with surface sulfurization, could effectively alleviate the diffluent polysulfides dissolution based on polar-polar interaction. Moreover, the confined polysulfides in the inner carbon shell could be fully catalyzed into Na₂S by atomic Co, leading to high S utilization. Therefore, the atomic Co in S@Co_n-HC plays a critical role in achieving sustainable cycling stability and high reversible capacity. By contrast, the intensive "shuttle effect" and incomplete sodiation reactions result in the inferior performance of the S@HC cathode. We have added this discussion into our revised manuscript.

Revised manuscript: "In order to guarantee the capacity reliability of the S@Co_n-HC cathode, the capacity contribution of S host, the Co_n-HC, was evaluated as well. The

Co_n-HC was fabricated from the S@Co_n-HC sample after dissolving the loaded S away by CS₂ solvent. The XRD results of Co_n-HC and S@Co_n-HC are shown in Fig. S20. It could be clearly seen that Co_n-HC does not show any S characteristic peaks, indicating that S has been completely removed. The discharge/charge profiles and cycling performance of Co_n-HC are shown in Fig. S21a, which displays a very low initial reversible capacity of 70 mAh g⁻¹, only retaining a reversible capacity of 40.1 mAh g⁻¹ after 200 cycles. By contrast, Fig. S21b clearly shows that the capacity contribution of Co_n-HC in the S@Co_n-HC cathode could be negligible. On the other hand, compared with S@HC, these atomic Co in S@Co_n-HC could effectively enhance the reversible capacity of the RT-Na/S@Co_n-HC batteries.”

“The schematic illustrations of electrode reaction mechanisms for the S@Co_n-HC and S@HC are shown in Fig. 6d and Fig. 6e. These atomic Co, with surface sulfurization, could effectively alleviate the diffusive polysulfides dissolution based on polar-polar interaction. Moreover, the confined polysulfides in the inner carbon shell could be fully catalyzed into Na₂S by atomic Co, leading to high S utilization. Therefore, the atomic Co in S@Co_n-HC plays a critical role in achieving sustainable cycling stability and high reversible capacity. By contrast, the intensive “shuttle effect” and incomplete sodiation reactions result in the inferior performance of the S@HC cathode.”

Figure 6 d, and e, Schematic illustrations of electrode reaction mechanism of S@Co_n-HC and S@HC.

Supporting information: “In order to make sure S host, the Co_n-HC, is inactive in the RT/Na-S batteries, we have studied the electrochemical performance of the plain matrix, in which S is removed from S@Co_n-HC to form Co_n-HC. The removing S

experimental process is following:

30 mg S@Co_n-HC was dispersed in 30 mL CS₂ solution. The mixture was ultrasonicated for 2 h, followed by washing by ethanol, acetone and DI water with several times, respectively. Finally, the obtained black powder was heated overnight in an oven at 80 °C for 12 h.”

Figure 5a, Cyclic voltammograms of S@Co_n-HC in RT-Na/S cell for 10 cycles within the voltage window of 0.8 - 2.8 V at a scan rate of 0.1 mV s⁻¹.

Figure S 18c. Cyclic voltammograms of S@HC in RT-Na/S cell for 10 cycles within the voltage window of 0.8 - 2.8 V at a scan rate of 0.1 mV s⁻¹.

Figure S20. XRD of $\text{Co}_n\text{-HC}$ and $\text{S@Co}_n\text{-HC}$.

Figure S21. **a**, discharge/charge curves and **b**, cycle performance of $\text{Co}_n\text{-HC}$ and $\text{S@Co}_n\text{-HC}$ at 100 mA g^{-1} .

References

- 1 Zhang, B.-W. *et al.* Platinum-Cobalt Bimetallic Nanoparticles with Pt Skin for Electro-Oxidation of Ethanol. *ACS Catalysis* **7**, 892-895, doi:10.1021/acscatal.6b03021 (2017).
- 2 Wang, D. *et al.* Structurally ordered intermetallic platinum-cobalt core-shell nanoparticles with enhanced activity and stability as oxygen reduction electrocatalysts. *Nat Mater* **12**, 81-87, doi:http://www.nature.com/nmat/journal/v12/n1/abs/nmat3458.html#suppleme

- ntary-information (2013).
- 3 Zhang, B. W., Yang, H. L., Wang, Y. X., Dou, S. X. & Liu, H. K. A Comprehensive Review on Controlling Surface Composition of Pt-Based Bimetallic Electrocatalysts. *Advanced Energy Materials* **0**, 1703597, doi:10.1002/aenm.201703597 (2018).
 - 4 Yang, X.-F. *et al.* Single-Atom Catalysts: A New Frontier in Heterogeneous Catalysis. *Accounts of chemical research* **46**, 1740-1748, doi:10.1021/ar300361m (2013).
 - 5 Qiao, B. *et al.* Single-atom catalysis of CO oxidation using Pt₁/FeO_x. *Nat Chem* **3**, 634-641, doi:http://www.nature.com/nchem/journal/v3/n8/abs/nchem.1095.html#supplementary-information (2011).
 - 6 Deng, D. *et al.* A single iron site confined in a graphene matrix for the catalytic oxidation of benzene at room temperature. *Science Advances* **1** (2015).
 - 7 Liu, P. *et al.* Photochemical route for synthesizing atomically dispersed palladium catalysts. *Science* **352**, 797 (2016).
 - 8 Han, Y. *et al.* Hollow N-Doped Carbon Spheres with Isolated Cobalt Single Atomic Sites: Superior Electrocatalysts for Oxygen Reduction. *Journal of the American Chemical Society*, doi:10.1021/jacs.7b10194 (2017).
 - 9 Liu, W. *et al.* Discriminating Catalytically Active Fe_Nx Species of Atomically Dispersed Fe–N–C Catalyst for Selective Oxidation of the C–H Bond. *Journal of the American Chemical Society* **139**, 10790-10798, doi:10.1021/jacs.7b05130 (2017).
 - 10 Liu, W. *et al.* Single-atom dispersed Co-N-C catalyst: structure identification and performance for hydrogenative coupling of nitroarenes. *Chemical Science* **7**, 5758-5764, doi:10.1039/C6SC02105K (2016).
 - 11 Yang, H. B. *et al.* Atomically dispersed Ni(i) as the active site for electrochemical CO₂ reduction. *Nature Energy* **3**, 140-147, doi:10.1038/s41560-017-0078-8 (2018).

- 12 Wang, Y. X. *et al.* Achieving High-Performance Room-Temperature Sodium-Sulfur Batteries With S@Interconnected Mesoporous Carbon Hollow Nanospheres. *J Am Chem Soc* **138**, 16576-16579, doi:10.1021/jacs.6b08685 (2016).

Reviewers' Comments:

Reviewer #1:

Remarks to the Author:

The authors have revised manuscript based on the reviewer's comments. Now this version of manuscript is suitable for publication.

Reviewer #2:

Remarks to the Author:

This manuscript has been thoroughly revised and can be published now.

Reviewer #3:

Remarks to the Author:

The revised manuscript has addressed most of my questions except for one, i.e., the XRD peak of dissolved polysulfides. In response to my question 5, the authors attribute the XRD peak at 22.97 degree to long-chain polysulfides. It is difficult to digest, that a dissolved species would have XRD peak, or in other words, the dissolved long-chain polysulfide species have the same crystalline packing order as S₈ molecules despite that they are dissolved in electrolyte. Do the authors suggest that these polysulfides are in a liquid-crystal state?

The revised manuscript has addressed most of my questions except for one, i.e., the XRD peak of dissolved polysulfides. In response to my question 5, the authors attribute the XRD peak at 22.97 degree to long-chain polysulfides. It is difficult to digest, that a dissolved species would have XRD peak, or in other words, the dissolved long-chain polysulfide species have the same crystalline packing order as S8 molecules despite that they are dissolved in electrolyte. Do the authors suggest that these polysulfides are in a liquid-crystal state?

Answer:

Thank you for your comment. We are sorry for the misunderstanding. We did not mean the dissolved species could have a XRD peak or in a liquid-crystal state. The XRD peak at 22.97 degree can be indexed to long-chain polysulfides, but it is from the polysulfides that confined in $\text{Co}_n\text{-HC}$ framework as solid state. As illustrated in Fig. 6d, the $\text{Co}_n\text{-HC}$ framework could effectively prevent the polysulfides to dissolve into electrolyte. Thus, the most of the long-chain polysulfides are confined in the inside of hollow carbon as solid state, which could be detected at 22.97 degree via operating *in-situ* synchrotron XRD in the RT-Na/S@ $\text{Co}_n\text{-HC}$ coin-cell device.

Figure 6 | d, Schematic illustrations of electrode reaction mechanism of S@Co_n-HC